



# High-resolution offshore wind resource assessment at turbine hub height with Sentinel-1 SAR data and machine learning

Louis de Montera[1], Henrick Berger[1], Romain Husson[1], Pascal Appelghem[2], Laurent Guerlou[1], Mauricio Fragoso[1]

[1]CLS Collecte Localisation Satellites, Ramonville-Saint-Agne, France
[2]Atmosky, Talence, France

*Correspondence to*: Romain Husson (rhusson@groupcls.com)

**Abstract.** This paper presents a method to calculate offshore wind power at turbine hub height from Sentinel-1 Synthetic Aperture Radar (SAR) data using machine learning. The method is tested in two 70 km x 70 km areas off the Dutch coast where Lidar measurements are available. Firstly, SAR winds at surface level are improved with a machine learning algorithm using geometrical characteristics of the sensor and parameters related to the atmospheric stability extracted from a high-resolution numerical model. The wind speed bias at 10m above sea level is reduced from -0.42 m s$^{-1}$ to 0.02 m s$^{-1}$ and its standard deviation from 1.41 m s$^{-1}$ to 0.98 m s$^{-1}$. After improvement, SAR surface winds are extrapolated at higher altitudes with a separate machine learning algorithm trained with the wind profiles measured by the Lidars. We show that, if profiling Lidars are available in the area of study, these two steps can be combined into a single one, in which the machine learning algorithm is trained directly at turbine hub height. Once the wind speed at turbine hub height is obtained, the extractible wind power is calculated using the method of the moments and a Weibull distribution. The results are given assuming an 8 MW turbine typical power curve. The accuracy of the wind power derived from SAR data is in the range ± 3 - 4% when compared with Lidars. Then, wind power maps at 200 m are presented and compared with the raw outputs of the numerical model at the same altitude. The maps based on SAR data have a much better level of detail, in particular regarding the coastal gradient. The new revealed patterns show differences with the numerical of as much as 10% in some locations. We conclude that SAR data combined with a high-resolution numerical model and machine learning techniques can improve the wind power estimation at turbine hub height, and thus provide useful insights for optimizing wind farm siting and risk management.

## 1 Introduction

Estimating the available offshore wind power at turbine hub height is a challenging problem due to the difficulty in measuring the wind profile in the boundary layer over the sea. Currently, two main methods are used to estimate the offshore extractible



power at hub height: numerical models and floating Lidars. Floating Lidars provide direct measurements of the complete wind profile at one location with a high temporal sampling, but they are very expensive to operate. Therefore, only one or two are typically used to sound large areas. Conversely, numerical models provide outputs over the entire area of interest, but they tend to flatten heterogeneities and extremes. Moreover, their errors are not precisely known, primarily because of the lack of representation of sub-grid processes. As a result, considerable uncertainty remains about actual offshore wind resources, which

can affect wind farm project planning and management. This is particularly relevant in coastal areas where processes are more complex and where the wind gradient is strong.

The need to improve wind speed assessment, and thus estimating more precisely wind power availability throughout wind farms' life cycle, has led to a growing interest in using remote sensing data to estimate wind resources (see, e.g., Hasager et al., 2015). Contrary to Lidars, spaceborne sensors have the advantage of sounding large areas with high spatial resolution.

However, they are not perfect: their revisit period is typically low (a couple of days for Sentinel-1 in Europe, for example), and they use indirect measurements by estimating the offshore surface wind from the sea state. Therefore, their measurements are impacted by several sources of potential error (sensor geometry, currents, algae, bright targets, bathymetry, turbulence, etc.). Moreover, the extrapolation of their measurements to hub height is not an easy task due to the variety of meteorological conditions.

Several studies have already attempted to assess offshore wind power potential with spaceborne scatterometers, such as ERS-1, ERS-2, NSCAT, QuickSCAT, and ASCAT (Sánchez et al., 2007; Pimenta et al., 2008; Karagali et al., 2014; Bentamy and Croize-Fillon, 2014; Remmers et al., 2019). However, the resolution of these instruments is at best 12.5 $km^2$, which is not adapted to coastal areas due to land contamination. In this context, Synthetic Aperture Radar (SAR) satellites are an interesting alternative because wind products derived from their measurements have a much finer resolution of 1 km. The potential of

SAR data has been assessed by numerous studies (Hasager et al., 2002; Hasager et al., 2005; Hasager et al., 2006; Christiansen et al., 2006; Hasager et al., 2011; Hasager et al., 2014; Chang et al., 2014; Chang et al., 2015; Hasager et al., 2020). However, validating SAR measurements with in-situ data has been limited (Ahsbahs et al., 2017; Badger et al., 2019; de Montera et al., 2020; Ahsbahs et al., 2020) and these studies concluded that important biases remained. Concerning the extrapolation, interesting methods have been proposed in the literature based on power laws or the statistical theory of turbulence (Grachev

and Fairall, 1996; Hsu et al., 1994; Badger et al., 2016); however, the problem has not been satisfactorily resolved and becomes increasingly critical as the typical height of windmills increases. Therefore, more research is needed to improve the estimation of wind resources at hub height with SAR data and convince the industry to use them.

In this study, we propose a method to overcome these limitations by using machine learning and by combining SAR data with a numerical model. On land, machine learning has been found to give good results compared to classical extrapolation methods

based on power laws or logarithmic laws (Türkan et al., 2016; Mohandes and Rehman, 2018; Vassallo et al., 2019). It has also been shown that, even if the algorithm is trained with a few in-situ instruments, it can be applied in a large area around them without significantly degrading the accuracy (Bodini and Optis, 2020; Optis et al., 2021). In this paper, we use SAR data from Sentinel-1 A and B satellites that provide the surface wind over the sea and improve them with the random forest algorithm.



First, a method is proposed to deal with the case where only surface in-situ measurements are available. In that case, the method requires two separate random forest algorithms: the first one improves SAR winds at surface level and the second one extrapolates them at turbine hub height. The reason for separating the method into two steps is that the improvement of SAR winds utilizes geometric properties of the sensor that are specific to the location. On the contrary, the algorithm extrapolating the surface wind to higher altitudes only depends on parameters related to the atmospheric stability. It can therefore be trained with Lidar data as a reference and applied in other areas. In the case where profiling Lidars are available in the study area study, these two algorithms can be combined into a single algorithm trained directly at hub height.

Both methods are tested in two areas off the Dutch coast where Lidar data are available. The SAR winds extrapolated at hub height are converted into a Weibull distribution and the extractible power is obtained by simulating the presence of a typical 8 MW wind turbine operating at 200 m. The resulting maps are presented and compared with the output of the numerical model in order to estimate the benefit of using these methods compared with a state-of-the-art technique.

## 2 Data and Methodology

### 2.1 High-resolution numerical model

The two zones of study are located off the Dutch coast and presented in Figure 1. They have an approximate size of 70 x 70 km. Their geographic extent was defined in order to include some offshore Lidars and a part of the coastline in order to observe the wind gradient. The WRF (Weather Research and Forecasting) Non-Hydrostatic Meso-scale Model developed by NOAA (National Oceanic and Atmospheric Administration) was run over these areas with a resolution of 1 km. This type of numerical model is representative of how wind resources are currently assessed by the industry. Here, it is used to estimate the atmospheric stability and extrapolate SAR data to turbine hub height. It was fueled at its boundary limits by a larger-scale model, the reanalyzed GFS (Global Forecast System) having a resolution of 0.5°. This larger-scale model was downscaled before using it to force the WRF model. The WRF model was run over a period from January 2015 to May 2020. It provides the wind speed and direction from the surface and up to 200 m, in increments of 20 m. It also provides other meteorological variables, such as air and sea surface temperature, surface heat flux, relative humidity and pressure.

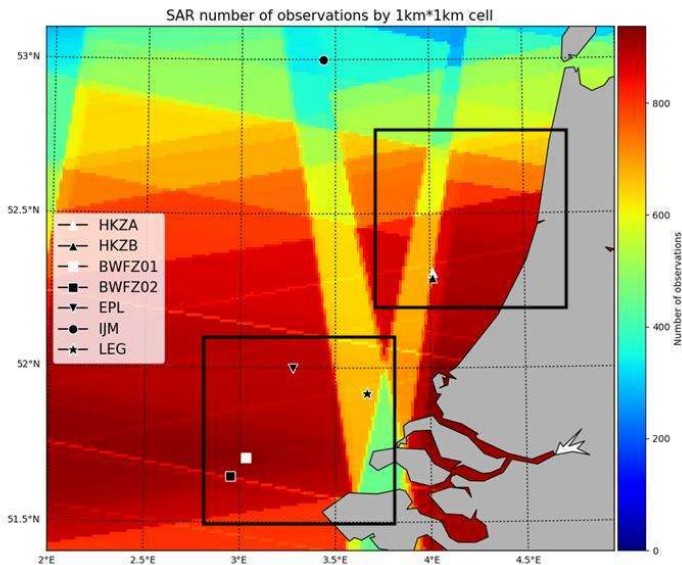

**Figure 1: Locations of the floating Lidars and total number of Sentinel-1 SAR L2 wind observations. The black boxes represent Zone 1 (bottom, latitude 51.50° - 52.09° / longitude 2.82° - 3.77°) and Zone 2 (top, latitude 52.15° – 52.74° / longitude 3.71° – 4.68°).**

### 2.2 In-situ instruments

The dataset comprises floating Lidars located off the Dutch coast (Figure 1). They were quality checked by our data provider C2WIND. The wind speed and direction are 10-minutes averaged around the observation times. There are 7 Lidars in the dataset, respectively named HKZA, HKZB, BWFZ01, BWFZ02, EPL, LEG and IJM. The vertical sampling and the duration of these Lidar measurements varies between observation campaigns and are displayed in Table 1. Zone 1 includes the Lidars BWFZ01, BWFZ02, EPL and LEG, and Zone 2 includes the Lidars HKZA and HKZB. The IJM lidar provides a very long period of measurements, but unfortunately it stopped operating before the availability of the Sentinel-1 B data, and therefore was not used. Similarly, the Lidar BWFZ02 was functioning only for 4 months and its small quantity of data was found to be unreliable.



| Lidar | Longitude | Latitude | First date | Last date | Number of levels | First altitude | Last altitude |
|---|---|---|---|---|---|---|---|
| HKZA | 4.011°E | 52.309°N | 2016-06-05 | 2018-06-05 | 11 | 30m | 200m |
| HKZB | 4.013°E | 52.292°N | 2016-06-05 | 2018-06-05 | 11 | 30m | 200m |
| LEG | 3.667°E | 51.917°N | 2014-11-17 | 2017-03-31 | 10 | 61m | 300m |
| EPL | 3.276°E | 51.998°N | 2016-05-30 | 2017-03-31 | 11 | 61m | 290m |
| IJM | 3.436°E | 52.998°N | 2011-11-02 | 2016-03-09 | 14 | 26m | 314m |
| BWFZ01 | 3.033°E | 51.71°N | 2015-06-11 | 2017-02-27 | 10 | 30m | 200m |
| BWFZ02 | 2.952°E | 51.65°N | 2016-02-12 | 2016-06-22 | 10 | 30m | 200m |

**Table 1: Main characteristics of the 7 floating lidars**


For each Lidar, the wind measured at the first altitude level is used to estimate the surface wind below at 10 m above sea level (a.s.l.), which is the altitude of SAR data. The extrapolation to 10 m is performed using a classical power law:

$$U_{10} = U_{min}\cdot\left(\frac{10}{Zmin}\right)^{\alpha} \hspace{4cm} \text{Eq. (1)}$$

where $U_{10}$ is the wind speed at 10 m in m s$^{-1}$, $U_{min}$ the Lidar wind speed at the first altitude level in m s$^{-1}$, and $Z_{min}$ the altitude of the first level in m, and $\alpha$ a non-dimensional exponent. Hsu et al. (1994) recommend choosing an exponent of 0.11 over the sea. We checked this hypothesis with HKZA and HKZB Lidars that were equipped with anemometers measuring wind speed
at 4 m a.s.l.. This power law was found to be correct on average. However, in order to refine the wind speed values at 10 m a.s.l., we adapted this exponent depending on the current atmospheric stability. The empirical exponents obtained with HKZA and HKZB Lidars were compared with the air-sea temperature difference provided by the high-resolution numerical model. The relation was fitted with a second-degree polynomial (Figure 2) and used to obtain the wind speed at 10 m a.s.l..

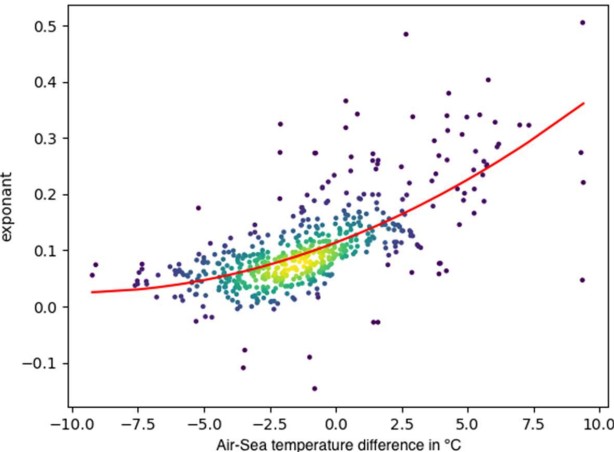


**Figure 2: Exponent of the power law between the wind speeds at 4 m and 40 m as a function of the air-sea temperature difference fitted with a second-degree polynomial fit (red curve).**

**2.3 Sentinel-1 SAR data**

Sentinel-1 A and B are two polar-orbiting satellites equipped with C-band SAR. This sensor, which records surface roughness, has the advantage of operating day and night at wavelengths not impeded by cloud cover. The Sentinel-1 Level 1 GRD (Ground Range Detected) product has a grid spacing of a few tens of meters, whereas the Level 2 wind products typically have a resolution of 1 km. The two satellites are located on the same orbit 180° apart and at an altitude close to 700 km. In Dutch

coastal waters, the acquisition mode is an Interferometric Wide (IW) swath using the TOPSAR technique, which provides a better-quality product by enhancing the image homogeneity (De Zan and Guarnieri, 2006). All Sentinel-1 A and B SAR images in IW acquisition mode from 2014 to 2020 in the study areas were collected. The total number of samples is shown in Figure 1, which shows that the coverage is not uniform.

The Level 1 images were calibrated and corrected from the instrument noise provided as metadata. A dedicated bright target

filtering was applied to remove Radar echoes created by ships, wind farms and other structures at sea. An additional filter (Koch, 2004) was used to identify heterogeneous signatures not related to wind, like currents, Radar interferences, and remaining bright targets. However, this filter has an increased sensitivity at low wind speeds, therefore, the identified pixels were not removed to avoid disrupting the wind speed Weibull distribution, which is necessary to estimate wind power. The information provided by this filter was only used to create maps of areas where wind power estimates are unreliable, typically

due to dense regions of wind turbines or mooring areas, which are well identified on average by the heterogeneity filtering. Then, Level 1 SAR products were degraded to a 1 km resolution and Level-2 surface winds at 10 m a.s.l. were obtained using a Bayesian inversion scheme using as inputs the wind speed obtained by inverting the SAR backscatter with the CMOD5.N geophysical model function (KNMI, 2008; ECMWF, 2008; Hersbach, 2010) and the outputs of ECMWF (European Centre for Medium-Range Weather Forecasts) NWP (Numerical Weather Prediction) model to constrain the wind direction. The

Level 2 product tiles were combined into a gridded map over the areas of interest, in order to form a data cube where each pixel corresponds to a time series of SAR measurements.

The revisit rate is one passage every two days, which occurs usually in the morning around 5 AM or in the evening around 5 PM (UTC). However, Figure 3, which gives the number of passages per year, shows that the constellation was only fully operational at the end of 2016. Therefore, we used SAR data from 2017, 2018 and 2019 when estimating the wind power to

ensure a complete and homogenous annual sampling. Figures 4 and 5 show the number of samples over these three years for each of the areas of interest.

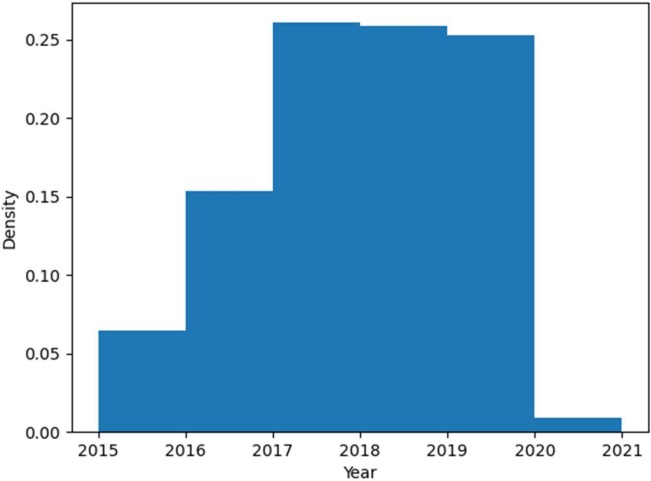

**Figure 3: Histogram of the number of SAR samples at Lidar HKZA's location. Before 2017, the constellation was not fully**
**operational. In 2020, only two months of data had been collected at the time of this study. Therefore, only 2017, 2018 and 2019 are complete with regular passes.**

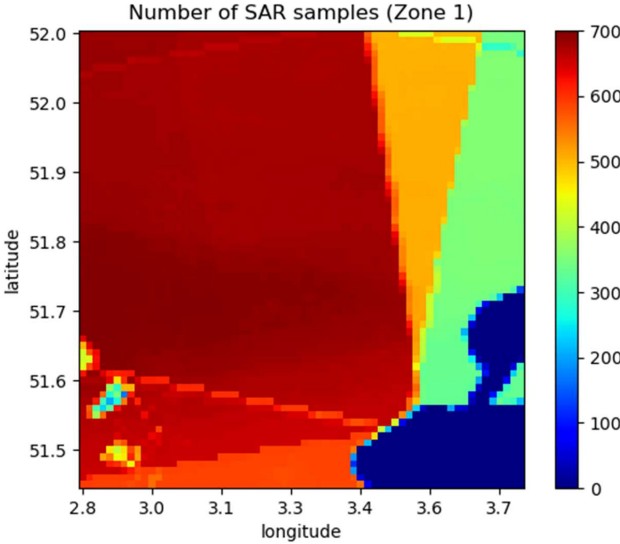

**Figure 4: Number of Sentinel-1 SAR wind samples available over Zone 1 during 2017, 2018 and 2019.**

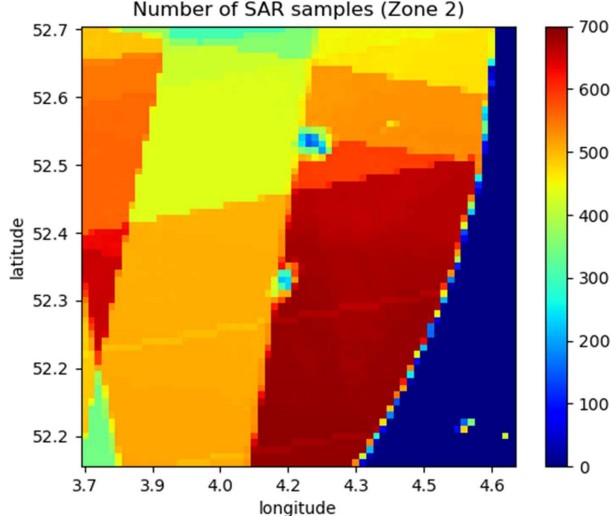


**Figure 5: Number of Sentinel-1 SAR wind samples available over Zone 2 in 2017, 2018 and 2019.**



**2.4 Wind power estimation**

The average extractible wind power $P$, hereafter called wind power, is calculated by multiplying point-by-point the wind speed
probability density function (pdf) by the power curve of a specific wind turbine, and then averaging the result. We chose to
simulate an 8MW turbine with a simplified power curve: 0 MW until a cut-in speed of 4 m s⁻¹, a linear increase until its nominal
output at 15 m s⁻¹, then a plateau at 8MW until 25 m s⁻¹, and a storm mode for higher values during which the turbine stops to
protect itself. A simple histogram could be used to estimate the wind speed pdf. However, due to the low number of SAR
samples, a more efficient technique consists in using the SAR data to fit a Weibull pdf, which usually describes the wind speed
accurately. The Weibull pdf is given by:

$$pdf(U) = \frac{k}{\lambda}\left(\frac{U}{\lambda}\right)^{k-1} e^{-(U/\lambda)^k}$$     Eq. (2)

where $\lambda$ is a scale parameter in m s⁻¹ and $k$ a dimensionless shape parameter. These parameters can be obtained by maximum
likelihood, or by the method of the moments with the following formulas (Pavia and O'Brien, 1986):

$$k = (\sigma/\mu)^{-1.086}$$     Eq. (3)

$$\lambda = \frac{\mu}{\Gamma\left(\frac{1}{k}+1\right)}$$     Eq. (4)


where $\mu$ is the mean wind speed, $\sigma$ the wind speed standard deviation, both in m s⁻¹, and $\Gamma$ the Gamma function.
The accuracy of these two methods was assessed with simulations. A random variable following a Weibull law with known
parameters was generated and the equivalent wind power computed. For both methods, the results were compared with the
original parameters. Figure 6 shows the mean absolute error as a function of the number of samples. With 500 samples, which
is approximately the amount of available SAR data available in the areas of interest (see Figures 4 and 5), the accuracy of these
methods is ± 2%. Both methods yield similar results, therefore the method of the moment, which is simpler and faster to run,
was chosen.

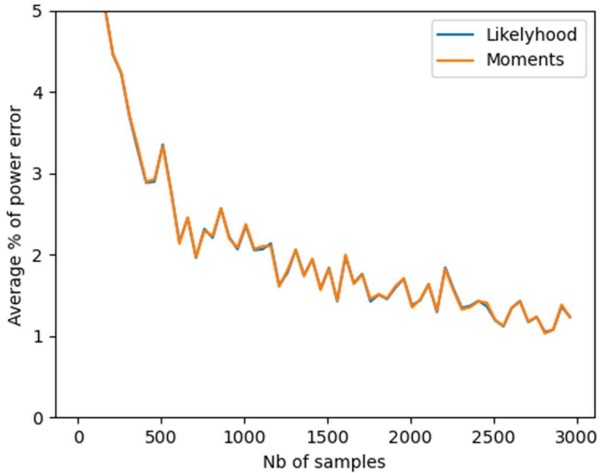

**Figure 6: Average absolute wind power error using the two estimation methods.**

### 2.5 Intra-diurnal variability

The main limitation of SAR data is their low temporal sampling (one passage every two days for Sentinel-1 in Europe). One advantage is that it guarantees the statistical independence of the measurements. However, the satellites are on a sun-synchronous orbit, which means that they pass always at the same time of the day. As a result, they cannot fully see the intra-diurnal variability of the wind. In order to assess this source of error, we simulated the satellites' passages by computing the mean wind speed using only Lidar measurements realized around 5 AM or 5 PM and compared it with the actual mean wind speed. The error was below 1%. The same analysis was done with the wind power and the error was found to be below 2%. Figure 7 shows the mean wind speed as a function of the hour of the day for each Lidar. It can be seen that the diurnal cycle is close to a sinusoid. Since the satellite passage times are separated by 12 h, this is enough to capture the majority of the 24 h period variability. Therefore, we conclude that intra-diurnal variability does not prevent the use of SAR data and that this source of error is limited. However, this conclusion needs to be validated in geographical areas where thermic winds are stronger.

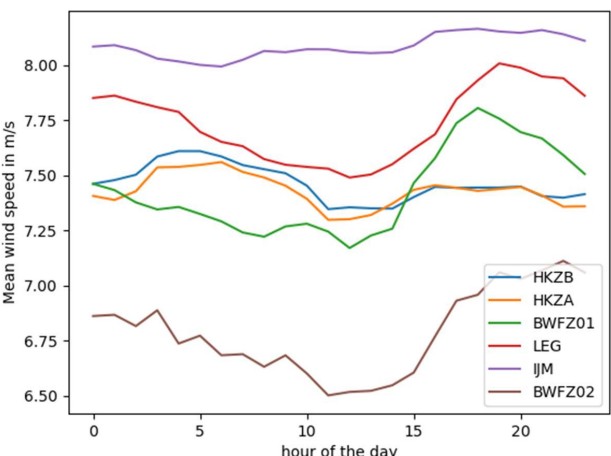

**Figure 7: Intra-diurnal variability of the mean wind speed for each Lidar.**

## 3 Improvement of SAR wind estimates and extrapolation at hub height

### 3.1 Machine learning at surface level

SAR surface winds are obtained by inverting the backscatter over a given pixel with a Geophysical Model Function (GMF). Originally, GMFs were designed to retrieve the wind from scatterometers. They were empirically designed using numerical models as a reference. However, SAR is a specific sensor and differences between the SAR and scatterometers backscatter in C-band may occur. In addition, numerical model outputs are not as reliable as in-situ data, especially in coastal areas. Moreover, GMFs may not fully capture the complex relation between the sea state and the wind, in particular because they assume a neutral atmosphere. As a result, SAR surface winds are typically biased when compared with in-situ buoys (see, e.g., de Montera et al., 2020). Therefore, it is necessary to improve the accuracy of the SAR winds obtained with a GMF. This is particularly important because the wind power is related to the cube of the wind speed, and therefore very sensitive to wind speed estimation errors. Given the complex relation between the sea state and the wind speed, and the number of factors able to influence it, machine learning was found to be an appropriate technique.

Two types of machine learning regressor were tested: the multi-layer perceptron and random forest. They were trained with the wind measured by the Lidars extrapolated to 10 m a.s.l. (the first Lidar level was extrapolated to this altitude with a power law, see Section 2.2). The input parameters were chosen by assessing their correlation with the error between the SAR and



Lidar measurements. The following parameters were found to have such a correlation: the SAR surface wind, the SAR wind direction, the azimuth angle, the incidence angle, the elevation angle, the backscatter, the thermal noise of the instrument and the difference between the azimuth angle and the wind direction (an important parameter in the inversion of the backscatter). In order to take the atmospheric stability into account, the air-sea temperature difference and the surface heat flux were extracted from the high-resolution numerical model and added as input parameters. The relative importance of these parameters

after the training stage is shown in Figure 8.

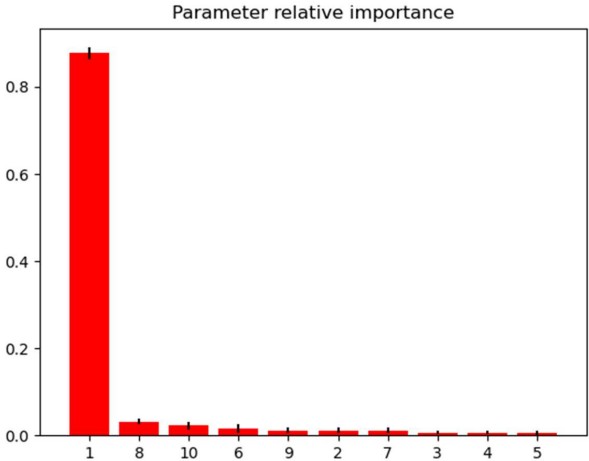

**Figure 8: Relative importance of the input parameters used to correct the SAR surface wind at 10m with the random forest algorithm (1 - SAR wind speed, 2 - SAR wind direction, 3 - azimuth angle, 4 - incidence angle, 5 - elevation angle, 6 -backscatter, 7 - thermal**
**noise, 8 - difference between the azimuth angle and the wind direction, 9 - air-sea temperature difference, 10 - heat flux).**

More than 1000 collocated data points between the Lidars and Sentinel-1 SAR could be found. The algorithm was trained with half of the data points, and the rest were used as a test dataset. Random forest was found to outperform neural networks in
terms of performance and training time. It is able to reduce the wind speed bias from -0.42 m s$^{-1}$ to 0.02 m s$^{-1}$ and its standard deviation from 1.41 m s$^{-1}$ to 0.98 m s$^{-1}$. Figures 9 and 10 show the errors between the SAR and the Lidars before and after machine learning. The bias is indeed reduced, and the cloud of points is thinner after machine learning. However, the resulting wind speeds are still biased at very low and very high wind speeds. These two ranges are more difficult to estimate because low wind speeds have little effect on the sea state, and because the relation between the sea state and the backscatter saturates
at high wind speeds. A multi-expert algorithm using three separate random forest algorithms to process respectively low, middle and high wind speeds was tested. However, this approach did not improve the results significantly.



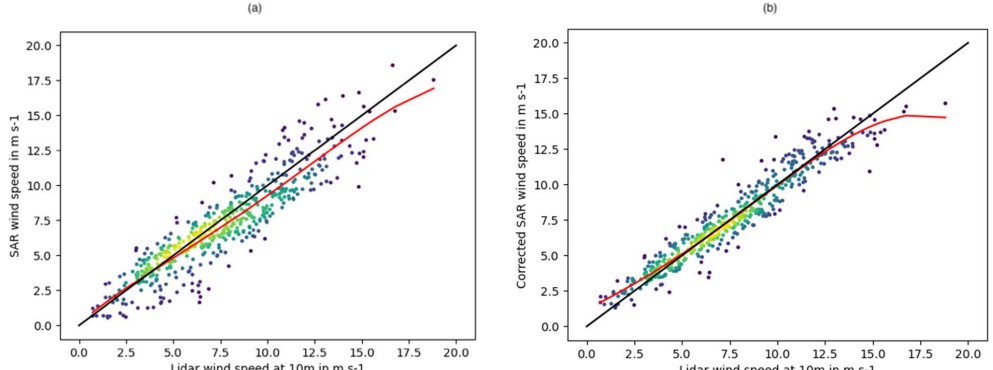

**Figure 9: Scatterplots between the SAR and Lidar wind speeds at 10 m (a) before machine learning (b) after machine learning, with**
**polynomial fits (red curves).**

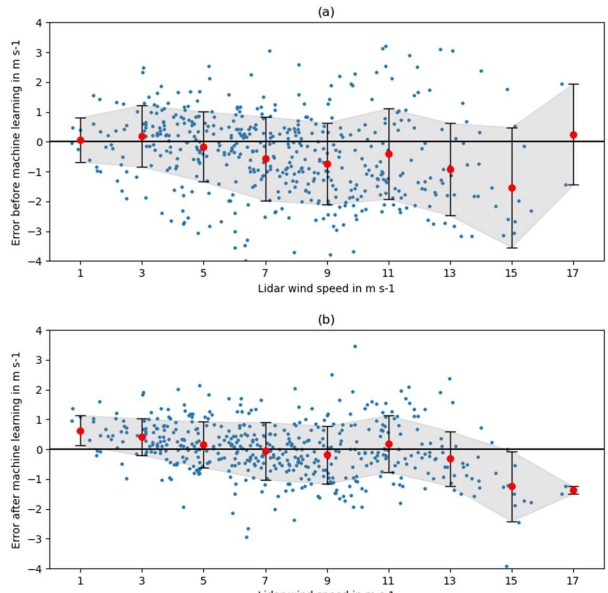

**Figure 10: Errors between the SAR and the Lidars as a function of Lidar wind speed at 10 m (a) before machine learning (b) after machine learning.**





### 3.2 Extrapolation at hub height


The extrapolation of surface wind speeds to higher altitudes is a challenging problem given the diversity of meteorological conditions and the variability of turbulence intensity in the boundary layer. Data analysis from offshore meteorological masts suggests that a simple power law could be sufficient to model the wind profile (Hsu et al., 1994). However, the analysis of our Lidar data shows that, above 40 m, this power law model is no longer accurate. This limitation has led some authors to use

numerical models to improve the extrapolation to higher altitudes (Badger et al., 2016). The advantage of numerical models is that they provide information about atmospheric stability through parameters like surface temperature and surface heat flux. In Badger et al. (2016), these surface parameters were averaged and combined with the similarity theory of Monin-Obukhov to extrapolate wind Weibull parameters. However, to our knowledge, this method was validated with only one meteorological mast in the Baltic Sea and not higher than an altitude of 100 m. Moreover, Optis et al. (2021) found that using machine learning

was more efficient than using a theoretical approach.

Therefore, we chose to extrapolate the instantaneous SAR wind speeds with machine learning using parameters extracted from the high-resolution WRF numerical model. The most relevant parameters were found to be the air-sea temperature difference and the surface heat flux. In order to increase the accuracy and adapt to the current meteorological conditions, the model ratio between the surface wind and the wind at hub height was also added. However, using comparisons to the Lidar measurements,

it was found that the numerical model outputs were less accurate in the lower boundary layer, and that the mean wind speed was biased below 40 m. Therefore, we decided to use the ratio between the wind speed at 40 m and higher altitudes. Comparing this ratio with the one found with the Lidar data showed that it is unbiased (for all Lidars, the bias was lower than 1%) and therefore suitable for extrapolating SAR winds. These parameters were used together with the corrected SAR winds at 10 m as input to a random forest algorithm and trained with 50% of the data points. The relative importance of the parameters is

shown in Figure 11. Figure 12 shows that the algorithm was successful in extrapolating wind speed because the bias compared to Lidars is stable with altitude and remains low and comprised within ± 3%. The mean wind speed error at 200 m against Lidars was -0.04 m s$^{-1}$ and its standard deviation 1.69 m s$^{-1}$. Thus, this method provides an almost unbiased estimate of the wind speed at hub height. We also attempted to follow the same approach as Badger et al. (2016), in which the extrapolation is performed on wind statistics. However, the extrapolation of the wind power with the corresponding ratio provided by the

numerical model was not as accurate as when the instantaneous winds were extrapolated first.

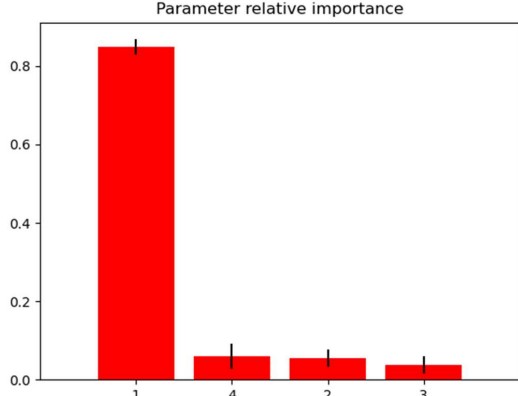

**Figure 11: Relative importance of the input parameters used to extrapolate the SAR surface wind to 200 m (1 - Corrected SAR wind speed, 2 - ratio of the numerical model wind speeds between 40 and 200 m, 3 - air-sea temperature difference, 4 - surface heat flux)**


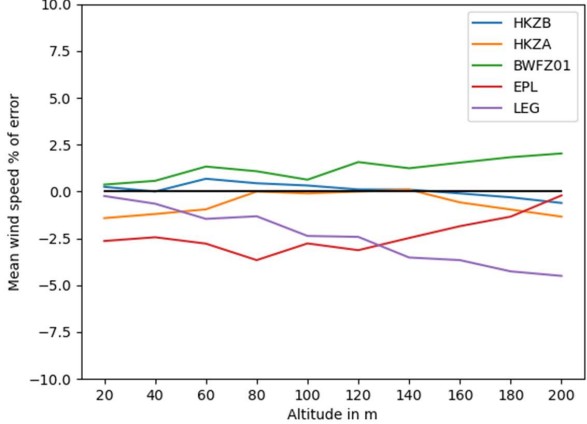

**Figure 12: Bias of the extrapolated SAR wind speed at each Lidar location.**




### 3.3 Machine learning at hub height with Lidar data

When profiling Lidars are available in the study area, the method accuracy can be improved by combining the correction of SAR surface winds and their extrapolation to higher altitudes into a single random forest algorithm. In that case, the algorithm is trained directly at hub height with all the input parameters together. The relative importance of the input parameters after

the training is shown in Figure 13. The results with the test dataset are shown in Figure 14. At 200m, the wind speed biases compared to each Lidars are within ± 2%. The total bias is 0.04 m s$^{-1}$ and the standard deviation 1.61 m s$^{-1}$. This result is better than the method presented in Section 3.2 using two separate steps. Therefore, this second method should be used if on-site profiling Lidars are available.

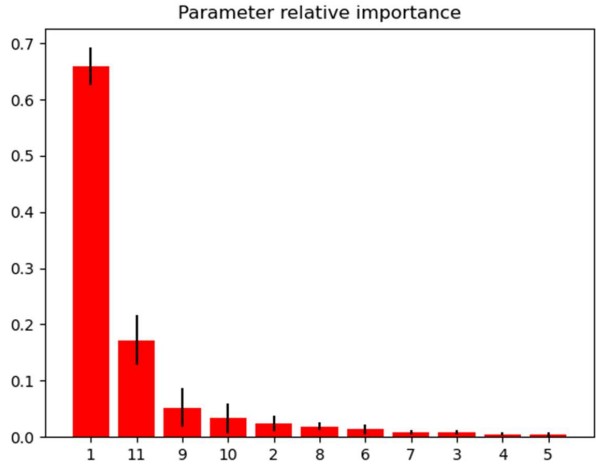


**Figure 13: Relative importance of the input parameters used to correct and extrapolate the SAR wind speed at 200m (1 - SAR wind speed, 2 - SAR wind direction, 3 - azimuth angle, 4 - incidence angle, 5 - elevation angle, 6 -backscatter, 7 - thermal noise, 8 - difference between the azimuth angle and the wind direction, 9 - ratio of the numerical model wind speeds between 40 and 200 m, 10 - air-sea temperature difference, 11 - heat flux).**

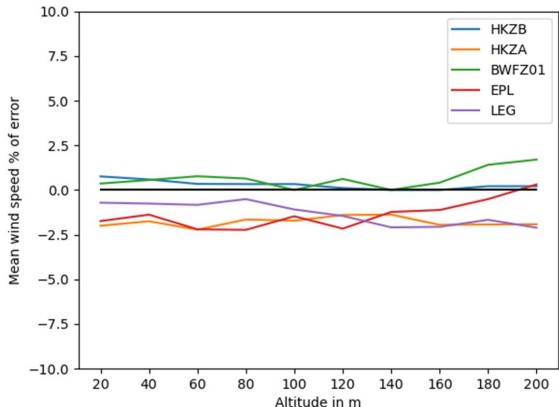


**Figure 14: Bias of the extrapolated SAR wind speed at each Lidar location (algorithm trained directly at hub height).**

## 4 Results

### 4.1 Correction of the standard deviation

As explained in Section 2.4, in order to estimate the extractible power, the Weibull parameters of the wind are needed. These parameters are directly linked to the first two moments of the wind speed distribution, which are the mean wind speed and the wind speed standard deviation (see Eqs. 3 and 4). Therefore, an accurate estimation of these two moments alone is enough to guarantee a low error of the extractible power. The extrapolation methods presented above provide an unbiased estimation of the mean wind speed. However, the wind speed standard deviation was found to be biased and underestimated in our case.

This occurs because machine learning estimates the most probable value of the wind speed, which does not necessarily reproduce the original distribution shape of the data. For example, at very low wind speeds, SAR sensors are often unable to detect any effect on the sea state. Therefore, in this range, machine learning tends to produce the same most probable value, regardless of the SAR wind speed. The same happens at very high wind speeds, for which the instrument saturates. As a result, the distribution tails become lighter, which reduces the wind speed standard deviation. Figure 15 shows an example of the

wind speed distribution obtained at 200 m after machine learning compared to the one obtained with Lidar HKZA. The standard deviation bias was on average - 6% and - 9% respectively for the two-step and single-step methods. Therefore, the opposite corrections were applied before computing the extractible power. In order to ensure a fair comparison with the numerical model, the same approach was applied to its outputs. In this case, was shifted to the left (Figure 15), which means that the model underestimates wind speed. This bias of - 4% was also corrected.



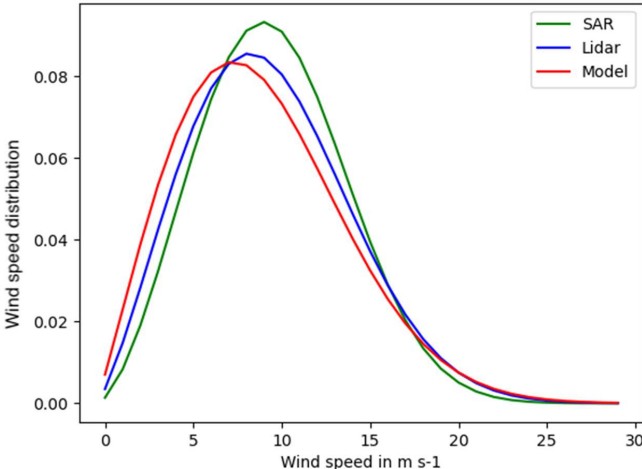


**Figure 15: Wind speed distribution at 200m measured by Lidar HKZA compared to the ones obtained with SAR data combined with machine learning at hub height, and the numerical model at the same location.**

### 4.2 Extractible wind power at hub height

Once the wind speed mean and standard deviation had been corrected for their biases, the estimation of the extractible power was done with the method presented in Section 2.4. The Weibull parameters were obtained with Eqs. 3 and 4, and then the wind power was obtained by multiplying the Weibull distribution (Eq. 2) by the typical 8 MW turbine power curve. Figures 16 and 17, respectively, show the results for the two-step method and the single-step method using profiling Lidars. The method accuracy is ± 4% in the first case and ± 3% in the second case. When on-site profiling Lidars are available, the accuracy

is close to the error bar of the wind power retrieval method (i.e., ± 2%, see Section 2.4). When profiling Lidars are not available and the two steps method must be used, our result indicates that the loss of accuracy would be limited. However, this conclusion needs to be confirmed in another geographical location where the correction of SAR surface winds could be performed with an algorithm trained with a different dataset such as metocean buoys, providing a higher number of collocated points, and where the extrapolation step would be validated with independent Lidar measurements. Due to the short distances between the

Lidars used in this study, such a validation could not be realized here. However, we also tested another extrapolation method without any machine learning by using directly the ratio given by the numerical model. In that case, the wind power error was comprised between ± 7%, which is accurate enough to provide useful high-resolution maps.

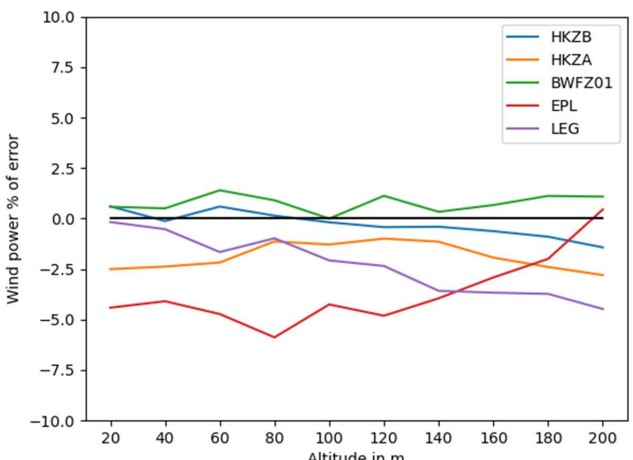

**Figure 16: Bias of the SAR extractible power at each Lidar location (two-step algorithm).**

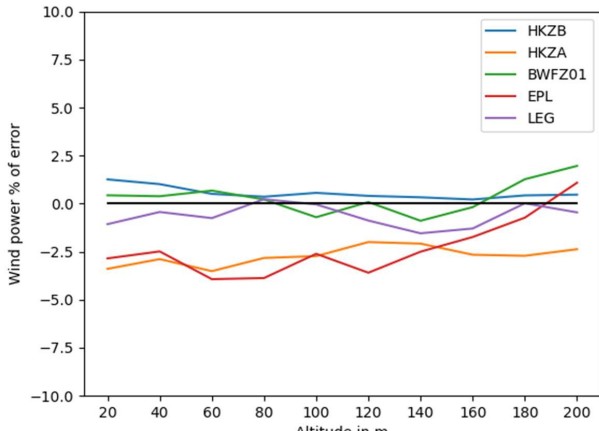

**Figure 17: Bias of the SAR extractible power at each Lidar location (algorithm trained directly at hub height).**



### 4.3 Wind power maps at hub height

Figures 18 and 19, respectively, show the wind power at 200 m over Zone 1 and Zone 2 predicted by the numerical model. Figures 20 and 21 show the wind power obtained from SAR data for the same areas using the algorithm trained directly at hub height. Figures 22 and 23 show the difference in percentage between the maps obtained with the numerical model and the ones obtained with the SAR. The use of SAR data significantly increases the level detail compared to the numerical model outputs. That correction can reach as much as 10% of the wind power between two sites separated by less than 20 km.

Some artefacts are still visible on the maps and need to be corrected in the future. For example, the swath edges can still be seen. Moreover, in some areas, the estimation was less reliable, mainly due to bright targets that could not be filtered. Some of these areas are linked to existing wind farms having a high density of turbines, while other areas had large numbers of stationary shipping vessels. The presence of these artefacts was measured by a Koch filter and a quality flag was created. Figures 24 and 25 show the percentage of data flagged as 'bad quality', and therefore the areas where the assessment is

unreliable. In addition, in Zone 1, a series of three unrealistic 'waves' can be seen close to the coast. We could check that these patterns correspond to similar 'waves' of sand in the seabed. The bathymetry in these shallow waters seems to affect currents and, therefore, the SAR backscatter.

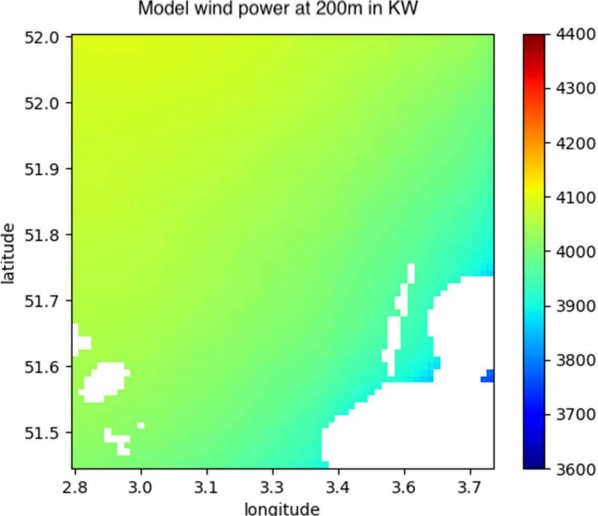

**Figure 18: Model wind power at 200m for a typical 8 MW wind turbine over Zone 1.**

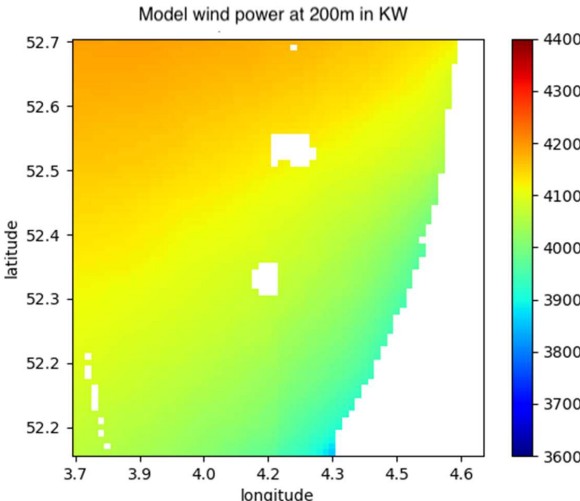

**Figure 19: Model wind power at 200m for a typical 8 MW wind turbine over Zone 2.**

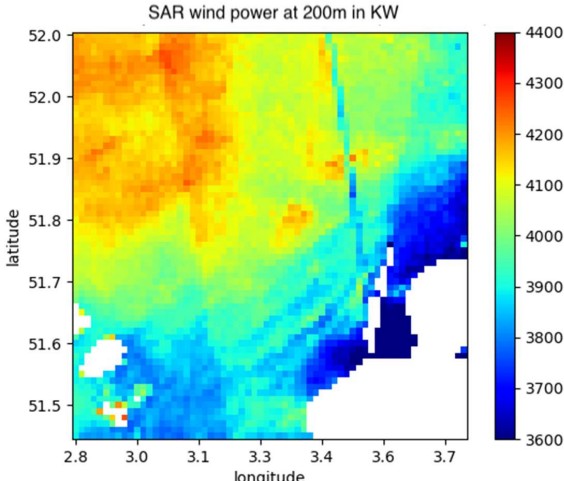

**Figure 20: SAR wind power at 200m for a typical 8 MW wind turbine over Zone 1 (algorithm trained directly at hub height).**

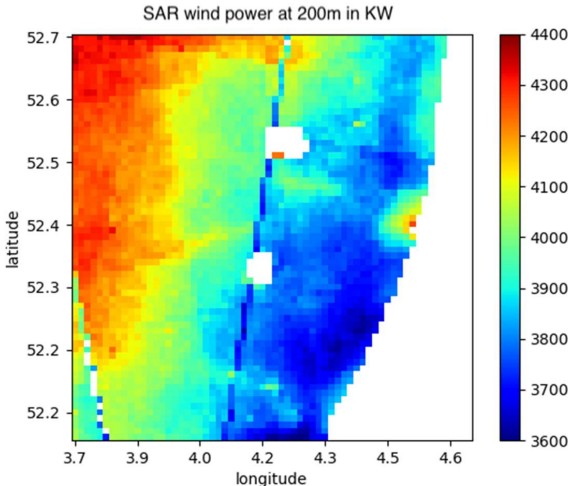


**Figure 21: SAR wind power at 200m for a typical 8MW wind turbine over Zone 2 (algorithm trained directly at hub height).**

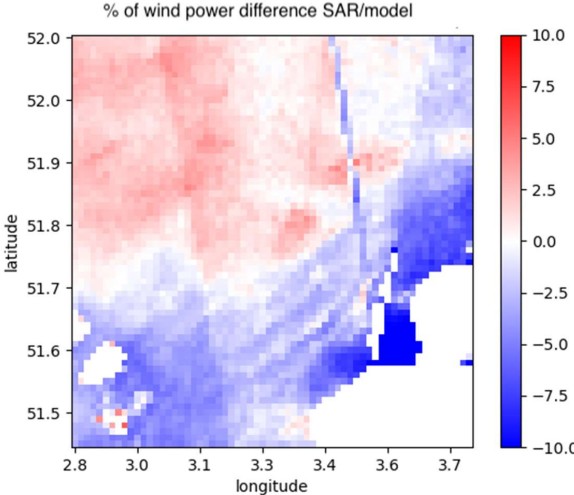

**Figure 22: Percentage of difference between SAR wind power and the model wind power over Zone 1.**

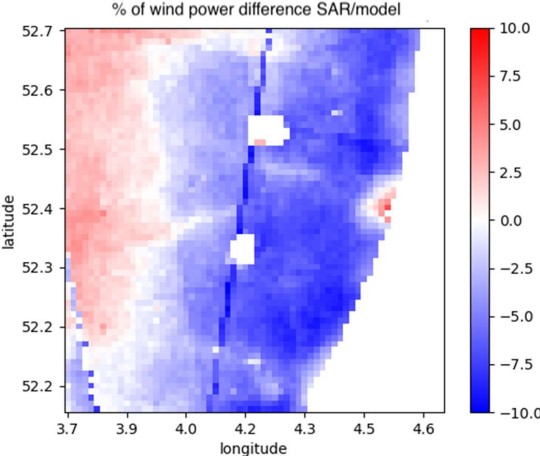

**Figure 23: Percentage of difference between SAR wind power and the model wind power over Zone 2.**

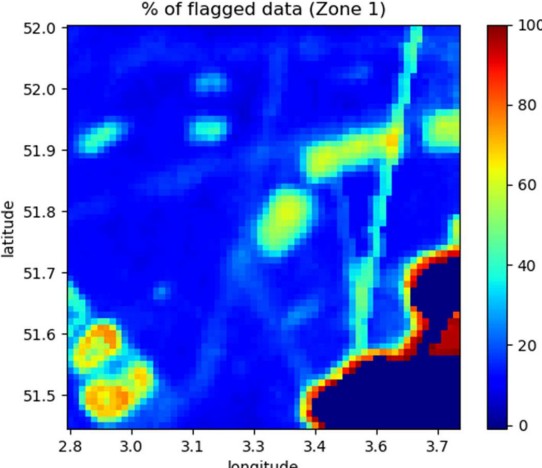

**Figure 24: Percentage of data flagged as 'bad quality' over Zone 1.**

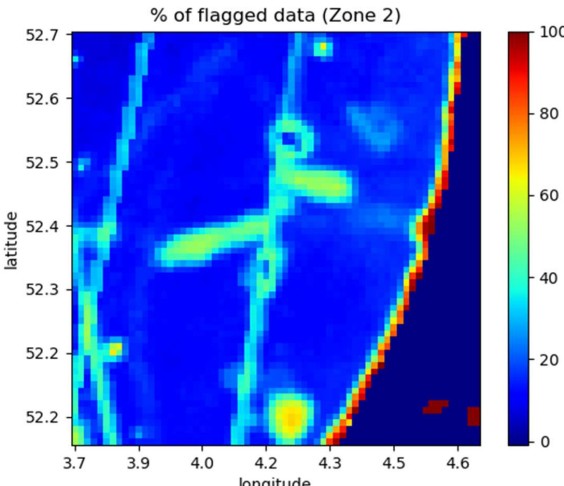

**Figure 25: Percentage of data flagged as 'bad quality' over Zone 2.**


### 5 Conclusion

A new method for estimating the extractible power at turbine hub height based on SAR data and machine learning has been presented. If profiling Lidars are available, the machine learning algorithm can be trained directly at turbine hub height with geometrical parameters of the SAR sensor and parameters related to the atmospheric stability. If no Lidar is available, the method can be separated into two steps: first correcting SAR surfaces winds with machine learning using surface wind measurements as a reference and then extrapolating these winds to higher altitudes with a second algorithm. The method was

tested in two areas off the Dutch coast using data from 5 Lidars. The extractible wind power maps were computed assuming a typical 8 MW turbine. At 200m above sea level, the accuracy of the method in which the algorithm is trained directly at hub height was 2% for the wind speed and 3% for the wind power. Regarding the two-step method, the accuracies were 3% and 4% respectively. One must add the error due to intra-diurnal variability, which was estimated to be less than 1% for mean wind speed and 2% for wind power in these areas.

Compared to the maps provided by the numerical model, this method has the advantage of providing a much higher level of details. In the areas affected by the coastal gradient, the difference between the SAR maps and the outputs of the numerical model can reach 10% of the wind power over short distances of less than 20 km. Therefore, using SAR data combined with a



high-resolution numerical model and processing them with machine learning can improve the assessment of the wind resource. It can provide useful insights to optimize wind farm siting and risk management.

Further research should focus on removing some artefacts remaining on the SAR maps, such as the swath edges, bright targets, and the effect of bathymetry. The method could also be improved by identifying other useful input parameters for machine learning, like the cross-polarization backscatter, which is more sensitive to strong winds. One objective is also to improve the machine learning algorithm in order to obtain a better description of the Weibull distributions tails and avoid having to adjust them with a reference distribution. Finally, the method needs to be generalized to other geographical areas with independent

in-situ measurements from Lidar and classical metocean buoys for assessing the wind speed accuracy at the sea surface and higher altitudes.

**Acknowledgments**

The authors would like to thank ESA for providing Sentinel-1 data and the Dutch Ministry of Economic Affairs and Climate Policy for providing the Lidar data. We would also like to thank Rémi Gandoin from C2WIND for checking the quality of the Lidar data and his useful insights, and Cynthia Johnson for correcting the spelling and improving the article style. This research was funded by the French Space Agency (Centre National d'Etudes Spatiales).

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
