# Peer review of "High-resolution offshore wind resource assessment at turbine hub height with Sentinel-1 SAR data and machine learning"

_Wind Energy Science, 2021_

## Referee Comment (RC1)

**High-resolution offshore wind resource assessment at turbine hub height with Sentinel-1 SAR data and machine learning**

*Louis de Montera, Henrick Berger, Romain Husson, Pascal Appelghem, Laurent Guerlou, Mauricio Fragoso*

**REVIEW**

**GENERAL COMMENT:**

The paper describes a machine learning (ML) approach to vertically extrapolate SAR-derived wind speeds in offshore domain. The problem addressed in the analysis is very relevant, and the solution proposed leverages state of the art techniques, and would be beneficial for the offshore wind energy community. However, at the current stage the quality of the presentation of the analysis is limited, and many more details are needed to fully understand and evaluate the procedure followed by the authors in the application of the machine learning models. Therefore, I invite the authors to consider my comments below, before I can recommend the paper for publication.

As a final note, this reviewer is not an expert in satellite retrievals, so this review mainly focuses on the atmospheric science and machine learning part of the study.

**MAJOR COMMENTS:**

1. In general, the main limitation of the current draft is the lack of scientific rigor in the presentation of the approaches used in the analysis. Please remember that you want to make your work replicable after one has read your paper. Below I have added several specific comments to provide examples of this issue. Please, re-consider your technical and statistical explanations and add details where needed. Also, the use of the word "error" and some other qualitative terms should be carefully revised, too.

2. The error quantification in Section 4.2 comes after the ML-derived wind speed distributions have been corrected based on hub-height observations in Section 4.1. If I got this right, this step does not make much sense from a scientific point of view. You are presenting error metrics for the ML-extrapolated winds after using hub-height observations to correct them: first, why should one even consider applying a ML-based approach to derive hub-height winds if observations are available? And then, the ML error quantification should NOT involve any correction using hub-height observations, as this would of course "artificially" improve the ML performance.

**SPECIFIC COMMENTS:**

1. Line 27: "floating lidars" are not a "method" to retrieve hub-height wind speed. Please rephrase.
2. L. 33: what type of "heterogeneities" are you referring to in the offshore wind context?
3. Please add references to the first and second paragraphs of the introduction (currently, the first paragraph has zero).
4. L. 38: a lidar IS a remote sensing instrument, too. Please rephrase to better convey your comparison between lidars and satellite-based retrievals.
5. Lines 65-70: please specify in which section you deal with each of the tasks you mention in the paragraphs.
6. Line 80: wasn't WRF developed at NCAR? Add a reference.
7. Please add a reference for GFS, too.
8. Did you run WRF in chunks (e.g., separate runs every month) or as a single run?
9. Figure 1: please make the fonts bigger (in the axes and the colorbar). Also, specify in the caption what the considered time period is. Also, please specify if latitude is N or S, and if longitude is E or W.
10. Section 2.1: which planetary boundary layer scheme did you use in WRF? Please clarify.
11. Throughout the paper, please consider using a different color scheme. The rainbow color has quite a lot of issues, see for example https://www.climate-lab-book.ac.uk/2014/end-of-the-rainbow/
12. Line 93: again, lidars are not in-situ instruments.
13. L. 95: please provide more details on the QC – is there a reference?
14. L. 100: if two lidars were not used in the analysis, I guess there is no need to include them in the map and in the text.
15. Table 1: "lowest" and "highest" instead of "first" and "last" when talking about altitudes.
16. Figure 2: typo on the y-axis label. Also, what do the shading of the dots represent?
17. Line 115: what do you mean by "correct on average"? Please provide quantitative assessments.
18. L. 147: what do you mean by "around 5 AM or 5 PM"? Please be specific and provide a plot/histogram/table if necessary.
19. I don't think Figure 3 is needed. In the text, you can simply state that you use data from 2017 to 2019 because the satellite constellation was not fully operational before that. No need to insert an histogram here (if you really want, please move it to the supplementary information).
20. What is the difference between the map in Figure 1 and what is shown in Figures 4 and 5? If it is just a matter of the years being plotted, I would suggest plotting the 2017-19 data in Figure 1 directly, so that Figures 4 and 5 can then be removed from the paper as they do not add much information.
21. Figure 6: "number" instead of "nb".  Also in the caption, please specify which methods you are using to provide more context.

22. L. 199: how do you quantify the "error"? What does "around" mean? Please be specific: you want to make your work replicable!
23. Figure 7: please specify the height at which wind speed is considered (y-axis label). Also, is time on the x-axis UTC or rather local time? Please specify.
24. L. 214: please provide context as well as references for the sentence "In addition, numerical model outputs are not as reliable as in-situ data, especially in coastal areas.". Also, what do you mean here with in-situ data? Once again, lidars are not.
25. L. 222: from the text, it is not immediately clear the purpose of these two ML models. This becomes clear later in the section, but please state that here too.
26. L.224: you need to define what "error" means for you with a precise statistical metric. Bias? RMSE? $R^2$? Or…?
27. L. 225: what do you mean by "such a correlation"? Please provide a threshold or a quantitative measure of what you did.
28. L. 226: please clarify what the azimuth angle, the incidence angle, the elevation angle, the backscatter are for a reader not familiar with satellite data retrievals.
29. Figures 8, 11, 13: please label the x-axis with actual names (rotated to get enough space), not numbers.
30. L. 239: how was the test set built? Was a random half of the data, or…? At all sites, or…?
31. L. 240: please provide additional details on the machine learning models and their training. Did you use cross-validation? What hyperparameters did you set? What is the structure of the neural network chosen? Did you train the model at all sites together, or at one site at a time?
32. L. 240: are you talking about mean bias? Please clarify.
33. Figure 9: again, specify what the different colors for the dots means. Also, larger fonts please. Also, is this for the test set only? At all sites? Please clarify.
34. Figure 10: "error" is too vague. Do you mean bias? Also, I would suggest mobbing this figure to the supplement, as all the information in it is already included in Figure 9, and it is not discussed in detail in the main text.
35. Section 3.2: the first paragraph could be moved to the introduction.
36. Section 3.2: once again, more details are needed to fully understand how the machine learning approach was applied. See my other specific comment above for specific questions that need to be addressed.
37. Figure 12: the y-axis label is not clear to me.
38. Do you have any explanations on why for some lidars in Fig. 12 the performance decreases with height, while for some others it actually increases?
39. Figure 15: please correct the y-axis label.
40. Section 4.1: it is not clear to me how the correction is performed. Please provide additional details to make your work replicable.
41. In Section 4.2, you state that "Due to the short distances between the Lidars used in this study, such a validation could not be realized here.". To me, lidars that are about 100 km from each other would still allow for a validation using one for training and another for testing.

**42.** Figg. 16, 17: the y-axis labels are not specific enough.
**43.** You can combine Figures 18 and 19 into a single one with two panels. Same for 20-21 and 22-23, and 24-25.
**44.** A data or code&data availability statement is missing.
**45.** A conflict of interest statement is missing.

---

## Author Comment (AC1)

**Manuscript WES-2021-35**

**Response to Anonymous Referee 1**

Dear Editor, dear Anonymous Referee,

First of all, we would like to thank the Anonymous Referee for the objective high-quality review of the manuscript. We think it will greatly improve the quality of the paper and we are grateful for the time spent on it.

We will provide below some answer to the Anonymous Referee remarks, before producing a revised version of the paper.

**MAJOR COMMENTS:**

**1. In general, the main limitation of the current draft is the lack of scientific rigor in the presentation of the approaches used in the analysis. Please remember that you want to make your work replicable after one has read your paper. Below I have added several specific comments to provide examples of this issue. Please, re-consider your technical and statistical explanations and add details where needed. Also, the use of the word "error" and some other qualitative terms should be carefully revised, too.**

Indeed, the work should be replicable, therefore, we will give more details on the machine learning in the revised version, including the full parametrization. This problem is not really a lack a scientific rigor, but rather than we did not give enough details. Moreover, we will define clearly in the beginning the statistical terms that we use, like 'errors' (error time series) and 'biais' (systematic error, mean error, MBE). We will also change several labels of the y-axis of some key figure that were confusing: we will use 'wind speed bias in %' instead of 'mean wind speed percentage of error'.

Due to ATMOSKY corporate constraints, we cannot provide the full parametrization of the WRF model. We think this is not a limitation since the numerical modelling is not the main purpose of this paper. It is just used as source of meteorological parameters related to the atmospheric stability. We will only provide the references that were used to choose the PBL scheme.

**2. The error quantification in Section 4.2 comes after the ML-derived wind speed distributions have been corrected based on hub-height observations in Section 4.1. If I got this right, this step does not make much sense from a scientific point of view. You are presenting error metrics for the ML-extrapolated winds after using hub-height observations to correct them: first, why should one even consider applying a ML-based approach to derive hub-height winds if observations are available? And then, the ML error quantification should NOT involve any correction using hub-height observations, as this would of course "artificially" improve the ML performance.**

It is of course not scientifically correct to use the Lidar data to do an 'a posteriori' correction, as the Anonymous Referee explains. However, it does not affect the validation of the machine learning algorithm itself. Indeed, the machine learning was designed to provide instantaneous wind speeds,

(not wind statistics) and was validated with independent data in Section 3, before the 'a posteriori' correction presented in section 4.2.

We did this 'a posteriori' correction because, in the framework of wind energy, we are interested in the wind distribution, and we noticed that the tails of the distribution were not reproduced correctly after the machine learning. Since this problem has intrinsic causes related to the sensor and the algorithm (i.e., the satellite does not detect low wind speeds and the random forest algorithm cannot extrapolate), we considered it was acceptable to do this 'a posteriori' correction since we expected it to be stable and the same everywhere.

However, as the Anonymous Referee mentions, this is can be seen as a weak point of the study. Therefore, we did a further analysis and found that this effect could be compensated by applying a specific linear model on the values in the distribution tails, instead of the random forest algorithm. In other words, we now combine two models: the random forest for typical values and a linear model for extreme values. This improvement will be presented in the revised paper and removes the need for an 'a posteriori' correction.

**SPECIFIC COMMENTS:**

**1. Line 27: "floating lidars" are not a "method" to retrieve hub-height wind speed. Please rephrase.**

ok

**2. L. 33: what type of "heterogeneities" are you referring to in the offshore wind context?**

The term "heterogeneities" is indeed not appropriate. We are referring to the limited capability to resolve small scale phenomena that the numerical model cannot 'see' due to the scale truncation, and therefore flattens. We propose to rephrase it as follows:

"Conversely, numerical models provide outputs over the entire area of interest, but they tend to flatten extremes and are not capable of resolving small scale phenomena due to their physics and resolution. "

**3. Please add references to the first and second paragraphs of the introduction (currently, the first paragraph has zero).**

The beginning of the first paragraph corresponds to general statements regarding common practices in the wind energy industry that we observe. We will add this reference for example: "Best Practices for the Validation of U.S. Offshore Wind Resource Models Mike Optis, Nicola Bodini, Mithu Debnath, and Paula Doubrawa, National Renewable Energy Laboratory."

Concerning the end of the first paragraph, fact that the uncertainties related to numerical models are unknown is also a very general statement. Even when a validation of numerical models with in-situ instruments is available, these results cannot theoretically be applied to other locations or periods, which leads the investors to look for real data. We do not have a specific reference for this limitation of numerical models, but we think it is well-known truth.

**4. L. 38: a lidar IS a remote sensing instrument, too. Please rephrase to better convey your comparison between lidars and satellite-based retrievals.**

We will rephrase this sentence more clearly to explain that lidars are remote sensing instruments.

However, from the point of view of this study, we prefer to call lidars simply 'in-situ' instruments in the rest of the paper, rather than 'remote sensing' instruments. This is due to the scales. For example, in the case of rain, a disdrometer is a remote sensing instrument using a beam, but we would say it is an in-situ when compared with a meteorological radar, because of the important difference of scales involved. Here, it is the same between the satellite and the Lidars.

**5. Lines 65-70: please specify in which section you deal with each of the tasks you mention in the paragraphs.**

ok

**6. Line 80: wasn't WRF developed at NCAR? Add a reference.**

ok

**7. Please add a reference for GFS, too.**

ok

**8. Did you run WRF in chunks (e.g., separate runs every month) or as a single run?**

The WRF was run in chunks of 24h.

**9. Figure 1: please make the fonts bigger (in the axes and the colorbar). Also, specify in the caption what the considered time period is. Also, please specify if latitude is N or S, and if longitude is E or W.**

ok

**10. Section 2.1: which planetary boundary layer scheme did you use in WRF? Please clarify.**

We used a specific planetary boundary layer scheme adapted to coastal environments. Although, as explained previously, we cannot provide a full list of the parameters, we will provide references that were used to do it, like:

M. Diallo, Y. Roustanand, E. Dupont. Extrapolating wind field at hub height from synthetic aperture radar (SAR) and ensemble of weather and research forecast (WRF) wind estimates. WindEurope Offshore 2019, 26-28 November 2019, Copenhague, Danemark

**11. Throughout the paper, please consider using a different color scheme. The rainbow color has quite a lot of issues, see for example https://www.climate-labbook.ac.uk/2014/end-of-the-rainbow/**

Ok, we will use more appropriate colobar, like nipy_spectral.

**12. Line 93: again, lidars are not in-situ instruments.**

See answer to point 4.

**13. L. 95: please provide more details on the QC – is there a reference?**

There is no reference. We will provide more details. For example, the minimum number of packets and availability percentage.

**14.L. 100: if two lidars were not used in the analysis, I guess there is no need to include them in the map and in the text.**

ok

**15.Table 1: "lowest" and "highest" instead of "first" and "last" when talking about altitudes.**

ok

**16.Figure 2: typo on the y-axis label. Also, what do the shading of the dots represent?**

The shading of the dote represent the density of dots that is sometimes difficult to see on scatter plots

**17.Line 115: what do you mean by "correct on average"? Please provide quantitative assessments.**

Ok. 'Correct on average' means that is the exponent of the power law is unbiased when considering all data (average error is 0). However, we decided to refine it depending on the atmospheric stability because it can fluctuate importantly between stable and unstable conditions. We could rephrase by saying: "correct on average but does not distinguishes between the different stability regimes: neutral, stable and instable".

**18.L. 147: what do you mean by "around 5 AM or 5 PM"? Please be specific and provide a plot/histogram/table if necessary.**

We will give details: "The satellites pass at 5 AM or 5 PM depending on the orbit orientation, descending or ascending, respectively. The exact acquisition time can also vary by plus or minus 30 mn depending on the incidence angle under which the region of interest is observed."

We do not think a histogram would be necessary here.

**19.I don't think Figure 3 is needed. In the text, you can simply state that you use data from 2017 to 2019 because the satellite constellation was not fully operational before that. No need to insert an histogram here (if you really want, please move it to the supplementary information).**

Ok we will suppress figure 3.

**20.What is the difference between the map in Figure 1 and what is shown in Figures 4 and 5? If it is just a matter of the years being plotted, I would suggest plotting the 2017-19 data in Figure 1 directly, so that Figures 4 and 5 can then be removed from the paper as they do not add much information.**

Ok. Yes, it would make the paper lighter.

**21.Figure 6: "number" instead of "nb". Also in the caption, please specify which methods you are using to provide more context.**

ok

**22.L. 199: how do you quantify the "error"? What does "around" mean? Please be specific: you want to make your work replicable!**

The 'error' L.199 relates to the 'mean wind speed' calculated in the previous sentence. We will just use 'bias' instead in the revised version and not 'mean wind speed error'.

**23.Figure 7: please specify the height at which wind speed is considered (y-axis label). Also, is time on the x-axis UTC or rather local time? Please specify.**

Ok. The time is UTC (which often equate to local time here because we are close to Greenwich).

**24.L. 214: please provide context as well as references for the sentence "In addition, numerical model outputs are not as reliable as in-situ data, especially in coastal areas.". Also, what do you mean here with in-situ data? Once again, lidars are not.**

Again, see the answer to comment 4 that explains why in this context we used the term 'in-situ' instrument for lidars. This makes it easier for the reader to clearly identify the satellite when we use term 'remote sensing' and avoid unnecessary confusion. As stated before, we will mention that Lidar are remote sensing instrument in the introduction and explain our choice.

Regarding numerical models, there is no debate that in-situ real data are more accurate. We do not think a reference is needed, because it is obvious that coastal processes involve smaller scales than the ones in open seas, thus increasing potential errors of the model due to the scale truncation. We will explain it in the revised version.

**25.L. 222: from the text, it is not immediately clear the purpose of these two ML models. This becomes clear later in the section, but please state that here too.**

ok

**26.L.224: you need to define what "error" means for you with a precise statistical metric. Bias? RMSE? R2? Or...?**

Error is a term that has different meaning in different sentences. It has a meaning only relatively to a specific value, therefore we cannot define it once for all. Here in particular, there might be a confusion because an 's' is missing. The correct sentence is : "the errors between the SAR and Lidar measurements", which seems clear to us.

**27.L. 225: what do you mean by "such a correlation"? Please provide a threshold or a quantitative measure of what you did.**

We did not use a criterion, but a simple visual observation of the scatter plots. We will say it was done with 'visual inspection'. We evaluate the parameter importance, so we check the accuracy of our choices afterwards and a visual inspection is therefore enough here.

**28.L. 226: please clarify what the azimuth angle, the incidence angle, the elevation angle, the backscatter are for a reader not familiar with satellite data retrievals.**

Ok.

**29.Figures 8, 11, 13: please label the x-axis with actual names (rotated to get enough space), not numbers. 30.L. 239: how was the test set built? Was a random half of the data, or...? At all sites, or...?**

Ok. The feature importance is obtained by the attribute `feature_importances_` of the random forest regressor using scikit-learn, after the training phase with 50% of the data (all lidars). We will give these details.

**31.L. 240: please provide additional details on the machine learning models and their training. Did you use cross-validation? What hyperparameters did you set? What is the structure of the neural network chosen? Did you train the model at all sites together, or at one site at a time?**

The models were trained at all sites together because the Lidars are not very far and located in areas with similar wind patterns. Moreover, since the lidar measurements were obtained during different campaigns with different instruments, we avoid obtaining a result related to instrument differences by doing so.

The hyperparameters were set using cross-validation, we will provide them in the revised version, as well as the neural networks MLP architecture. The hyperparameters found are not very different from the scikit-learn default ones for the random forest regressor.

**32.L. 240: are you talking about mean bias? Please clarify.**

We use the term 'bias' as it is commonly used, meaning a systematic error (the mean of the errors time series, or mean error). In order to avoid confusion, we will add that it is also sometimes called Mean Bias Error (MBE) by other authors and define it clearly in the beginning.

**33.Figure 9: again, specify what the different colors for the dots means. Also, larger fonts please. Also, is this for the test set only? At all sites? Please clarify.**

Ok, yes this is the test set. For the colors, again, this is just the density.

**34.Figure 10: "error" is too vague. Do you mean bias? Also, I would suggest mobbing this figure to the supplement, as all the information in it is already included in Figure 9, and it is not discussed in detail in the main text.**

The term 'errors' used here does not seem vague to us since we explain it is the errors between the SAR and the Lidars. However, we will add that the red dots represent the 'bias' if it is not obvious (the mean of these errors).

We could move this figure to the supplements.

**35.Section 3.2: the first paragraph could be moved to the introduction.**

ok

**36.Section 3.2: once again, more details are needed to fully understand how the machine learning approach was applied. See my other specific comment above for specific questions that need to be addressed.**

Here too, we will provide the hyperparameters.

**37.Figure 12: the y-axis label is not clear to me.**

The label is indeed confusing. The y-axis is the percentage of bias compared to the mean wind speed. We will clarify and use 'wind speed bias (%)'.

**38.Do you have any explanations on why for some lidars in Fig. 12 the performance decreases with height, while for some others it actually increases?**

Actually, the performance does not increase or decrease with altitude depending on the lidar. As explained just above, the y axis is the bias in %, which can be positive or negative. The dispersion that can be observed when the altitude is increasing is just the error bar of our method that slightly

increases with altitude, as it could be expected. When the altitude increases, the bias of our method for a given lidar can be found in a larger interval around 0.

**39.Figure 15: please correct the y-axis label.**

Ok (we assume you are talking about Figure 14)

**40.Section 4.1: it is not clear to me how the correction is performed. Please provide additional details to make your work replicable.**

See the answer to major comment 2. We will not do this correction in the revised paper, but improve the method by merging the outputs of two algorithms depending on the wind speed range.

**41.In Section 4.2, you state that "Due to the short distances between the Lidars used in this study, such a validation could not be realized here.". To me, lidars that are about 100 km from each other would still allow for a validation using one for training and another for testing**

The maximum distance between the lidars is about 80 km. This would be enough on land, but on the seas, we have reservations because of the homogeneity of the wea surface and of the wind field. Moreover, there is the risk that our results could be more related to the different types of lidars that were used, rather than the geographic effect. We hope to publish soon a test of the method over an area located in another sea or ocean.

**42.Figg. 16, 17: the y-axis labels are not specific enough.**

ok

**43.You can combine Figures 18 and 19 into a single one with two panels. Same for 20-21 and 22-23, and 24-25.**

ok

**44.A data or code&data availability statement is missing.**

The data are available online (SAR image from ESA and Lidar data from RVO). However, due to the corporate constraints we have, unfortunately, we cannot provide our code.

**45.A conflict of interest statement is missing**

We have no conflict of interest

Best regards, the authors.

22/06/2021

---

## Author Comment (AC2)

Dear referee 2, please find the answer to your comments below:

***Review of manuscript "High-resolution offshore wind resource assessment at turbine hub height with Sentinel-1 SAR data machine learning" by Louis de Montera et al.***

*As I am providing the second review of the manuscript, and have had the change to go through the first reviewer's comments (referred to as RC1), I can start with pointing out that I fully agree with the raised criticism and recommended changes for an improvement of the manuscript.*

*Below I list my own major and minor comments, some of them being a repetition of those in RC1 but also including some additional input:*

*(comments in order of appearance in manuscript)*

*[l 11] When referring to "Lidar measurements", first time here in the abstract, please specify what kind of measurements you mean explicitly – e.g. Doppler wind lidar, somewhat ground-based, measurements of wind velocity profile in range relevant for wind energy applications, or similar.ll Please check the overall manuscript for a sufficiently specific terminology with this respect.*

We could write: "Doppler wind Lidar measurements from commercial LiDAR units deployed for offshore Wind Energy site investigations purposes"

*[ll 13-14] When stating a bias, you also need to mention the considered reference – please add this here.*

We will say it is the bias compared to Lidar measurements.

*[ll 19-20] When reading the sentence "The accuracy of the wind power…" it becomes not clear how you get to the numbers you compare. You should also refer to the process of deriving a power curve from (any) wind data, i.e. the involved derivation.*

We will make it clear that we multiply the Weibull distribution by the power curve of a typical turbine in order to obtain the extractible power.

*I am also wondering if you really need to consider this (as I understood later, very simplified power curved derivation) for your study, or instead could focus on a derivation of wind power density.*

The problem with deriving a total wind power density is that we would have a much higher error in the end when comparing with Lidars. This would lead to conclude that SAR satellites are not a good way to estimate wind resources, whereas, in practice, when multiplying the Weibull pdf by a power curve, the error is much lower. This is why we chose to use a power curve (we can explain that in details in the revised paper).

More precisely, the error is higher when computing the total wind power density, because one has to consider very low wind speeds and very high winds speeds. In these ranges, wind speed is difficult to measure precisely with SAR satellites and the machine learning is failing to produce accurate estimates due to the lack of sufficient training data. Our opinion is that, in these ranges, the turbine is usually not functioning anyway, so that there is no need to consider them in practice when looking at the extractible power with a power curve. So, to sum up, we chose this approach because it is closer to industrial applications and because it is more appropriate to test the use of SAR satellites. Estimating the total wind power density with SAR satellites seems however possible, but with more time and research in the future.

*[l 39] I believe you should refer here to ground- (or bottom-) based Lidars.*

ok

*[ll 43-44] The sentence seems incomplete – add "… of meteorological conditions [that may impact this extrapolation]", or similar.*

ok

*[l 59] Be more specific here: "found to give good results" for what explicitly?*

The correct reference was put in the wrong sentence 3 lines after this sentence. The reference is Optis et al., 2021 in which the authors compare machine learning and theoretical approaches to extrapolate the surface wind to higher altitudes. They conclude that machine learning is giving the best accuracy.

*[l 89] Here and at other places where you introduce already available models/methods, please add a reference – in this case, for WRF.*

We will give this reference:
NCAR/TN–556+STR
NCAR TECHNICAL NOTE
March 2019
A Description of the Advanced Research WRF Model Version 4

*[Figure 1] I suggest to add a larger map to help locating this cut-out. Please also introduce the used abbreviations.*

ok

*[Figure 1 and Table 1] Are you sure that all these datasets are from floating lidars? I have at least some doubt with respect to LEG and IJM. Please re-confirm.*

After checking, we confirm LEG, EPL and IJM are not floating Lidars but actually on platforms.

*[l 115] From own experiences and also in line with available recommended practices, I would not fully trust the lower height (as 4 m a.s.l.) measurements from floating lidar systems – mostly from in-situ sensors heavily influenced by the structure itself. Please have some thoughts on this, and possibly consider some added uncertainty.*

Ok. Anyway, we had time to improve our approach. In the revised paper, we will avoid extrapolating Lidars measurements from 40m to 10m using these unprecise buoys. Instead, we will do the geometric correction of SAR measurements with the US NDBC buoy networks (around 40 selected buoys measuring wind speed at 4m asl).

This will also clarify why we present two different methods: the advantage of the NDBC dataset is that there is a wide variety of azimuth and incidence angles in the training dataset. This training can therefore be applied anywhere, contrary to the second method, which is more precise, but requires on-site Lidars.

*[Figure 3] This figure is not very informative – please review the design and information included. Overall, I think you can and should reduce the number of figures in the manuscript, possibly combining some of them.*

OK. Figure 3 will be removed because it is not useful.

*[Figure 4 and Figure 5] In these plots it may be helpful to have the coastline (or something else) as reference.*

These figures will be removed and included in Figure 1 that will represent only 3 years of Sentinel1 data

*[section 2.4] As pointed out above, I think, the used power curve is too simplified. I would suggest to either apply a more realistic power curve, or instead consider another quantity as e.g. wind power density for this investigation.*

We will use a more realistic power curve:

https://nrel.github.io/turbine-models/DTU_10MW_178_RWT_v1.html

As explained above, we prefer not to compute the wind power density instead since the SAR is less precise in measuring low and high wind speeds.

*[Figure 6] I do not think that this figure is really needed, instead you could add more details in the text.*

Ok

*[section 2.5] I am confused by your mentioning of "one passage every two days" and "passage times are separated by 12 h" – please be more specific here.*

Ok. Yes, this is confusing. The satellite passes every two days, and this can occur at specific times: or in the morning or in the evening. The two possible time have a difference of 12h.

*[l 222] As already stated above, please add reference for the applied methods – here the "two types of machine learning regressor[s]".*

Ok

*[l 229] Please specify how "the relative importance" is defined and derived.*

Ok. The relative importance is based on the mean decrease in impurity (over the trees). We will give the python function in the text (`feature_importances_` attributes of random forest regressors in scikit-learn package)

*[l 239] Also the statement "Random forest was found to outperform neural networks" needs more explanation / details.*

The accuracy of the outputs of neural networks was found to be lower and the training time was much higher. We attribute the success of random forest to its ability to model non-linear relations in an easier way than neural networks, because it uses thresholds rather that activation functions.

*[Figure 9] Add a legend to the plots and the details of the red curve (fitting parameters).*

Ok. We will indicate the number of degrees of the polynomial fit and add a legend. We will also say the color is the density.

*[l 266] Again, "machine learning" needs more explanation and details.*

We will add it is a random forest regressor and give the parameters.

*[Figure 12 (and Figure 14)] It is not clear to me, why you have not used a smaller range for the vertical axis – please revise.*

We could use a -5% to +5% axis.

*[l 323] Sentence "In this case, …" is incomplete.*

The word 'the distribution' is missing.

*[Figure 18 and following] Please re-arrange these plots for better comparability – combine several plots in one figure, for instance.*

We could make 2 figures, one for each zone, with four subplots each (model wind power, Sar wind power, difference in percentage and quality of the data)

We would like to thank you for these helpful comments and interesting insights.

Best regards,

The authors

---

## Author Response (AR1)

**Revised version of manuscript WES-2021-35: answer to comments**

Dear Editor,

We finalized the revised version of WES-2021-35 and would like to draw your attention to the following points:

- Regarding Anonymous Referee 1 first major comment (replicability), we gave more details about the algorithms, the data, and the numerical model. We also improved the clarity of the paper and the definition of the statistical criteria. Most of these changes are explained in the answers to Anonymous Referee 1 minor comments related to this first major comment.

- We decided to present only the two-step method (a first algorithm correcting the SAR surface wind speeds and a second algorithm extrapolating these wind speeds to hub height) and renounced to present the method combining these two steps into a single one using on-site Lidar measurements. The first reason is that it makes the paper easier to read and reduces the number of figures, as recommended by both Anonymous Referees. The second reason is that the method combining the two algorithms is unapplicable in practice: typically, only one or two Lidars per site are installed for a short period to assess the resource, which is insufficient to create a training dataset.

- Regarding Anonymous Referee 1 second major comment about the 'a posteriori' correction of the wind speed standard deviation, we could identify the source of the problem. The wind speed standard deviation is underestimated because the machine learning algorithms are designed to provide only expected values. Therefore, they remove some variability related to the errors. In order to correct this effect and maintain the original variability of the wind speed, we artificially added a random variable having the same distribution as the errors. This appears to solve the problem and no 'a posteriori' correction is needed anymore.

Again, we would like to thank you and the Anonymous Referees for your contributions. Please find the detailed answers below.

Best regards,

The authors.

**Response to Anonymous Referee 1**

**MAJOR COMMENTS:**

**1. In general, the main limitation of the current draft is the lack of scientific rigor in the presentation of the approaches used in the analysis. Please remember that you want to make your work replicable after one has read your paper. Below I have added several specific comments to provide examples of this issue. Please, re-consider your technical and statistical explanations and add details where needed. Also, the use of the word "error" and some other qualitative terms should be carefully revised, too.**

We gave more details on the machine learning in the revised version, including the full hyperparametrization. We also gave more details about the data and the numerical model. However, due to ATMOSKY corporate constraints, we cannot provide the full PBL parametrization of the WRF model. We think this is not a limitation since it is just used as source of meteorological parameters related to the atmospheric stability and is not the main purpose of this paper. However, we provided the references that were used to choose the PBL scheme.

Regarding the term 'error', we clearly explained to which difference it refers each time it is used.

**2. The error quantification in Section 4.2 comes after the ML-derived wind speed distributions have been corrected based on hub-height observations in Section 4.1. If I got this right, this step does not make much sense from a scientific point of view. You are presenting error metrics for the ML-extrapolated winds after using hub-height observations to correct them: first, why should one even consider applying a ML-based approach to derive hub-height winds if observations are available? And then, the ML error quantification should NOT involve any correction using hub-height observations, as this would of course "artificially" improve the ML performance.**

We found out that the main reason why the wind speed standard deviation is underestimated. It is because the machine learning algorithm provides the expected value of the wind speed, which removes a part of its variability related to the errors. This does not affect the mean wind speed but squeezes its distribution. In order to compensate for that effect, we added a random variable having the same distribution as the error to the algorithm outputs. For each data point, we created 5 artificial output datapoints by adding different realizations of this random variable mimicking the errors. This helps the wind speed standard deviation to converge towards its definitive value. This method leads to an almost unbiased wind speed standard deviation and no 'a posteriori' correction is needed.

**SPECIFIC COMMENTS:**

**1. Line 27: "floating lidars" are not a "method" to retrieve hub-height wind speed. Please rephrase.**

L. 32: Currently, it is estimated by using numerical models and/or Doppler wind Lidars installed at the sea surface pointing upwards (NREL, 2020).

**2. L. 33: what type of "heterogeneities" are you referring to in the offshore wind context?**

We are referring to the limited capability of numerical models to resolve small scale phenomena due to their scale truncation. Moreover, the effect of this scale truncation is also unknown because Navier-Stokes equation are chaotic and unsolved. As a consequence, there is no theoretical formula for the error of numerical models and in-situ validations cannot be extrapolated to other locations or periods.

L. 35: Conversely, numerical models provide outputs over the entire area of interest. However, they are not capable of resolving small scale phenomena due to their physics and resolution. As a result, their errors are not precisely known and may vary in time and space. This is particularly problematic in coastal areas where processes are more complex and involve smaller scales.

**3. Please add references to the first and second paragraphs of the introduction (currently, the first paragraph has zero).**

L. 32: Currently, it is estimated by using numerical models and/or Doppler wind Lidars installed at the sea surface pointing upwards (NREL, 2020).

NREL. Best Practices for the Validation of U.S. Offshore Wind Resource Models. Optis, M., Bodini, N., Debnath, M., and Doubrawa, P., 2020, https://www.nrel.gov/docs/fy21osti/78375.pdf, last accessed 24 August 2021.

**4. L. 38: a lidar IS a remote sensing instrument, too. Please rephrase to better convey your comparison between lidars and satellite-based retrievals.**

From the point of view of this study, we prefer to call lidars simply 'in-situ' instruments, rather than 'remote sensing' instruments. This is due to the scales. For example, in the case of rain, a disdrometer is a remote sensing instrument using a beam, but we would say it is an in-situ when compared with a meteorological radar, because of the important difference of scales involved. Here, it is the same between the satellite and the Lidars.

L. 56: (in the context of this study, the term 'in-situ instruments' includes profiling Lidars, although technically they use remote sensing)

**5. Lines 65-70: please specify in which section you deal with each of the tasks you mention in the paragraphs.**

L. 79: Section 2 describes the SAR data used in this study, the numerical model, the Lidar data used as a reference to train the algorithms, and the formulas used to compute the wind power. Section 3 describes the two machine learning algorithms designed to improve the accuracy of SAR surface winds and extrapolate them to hub height, respectively. The reason for separating the method into two algorithms is the scarcity of offshore Lidar data. Since the first algorithm correcting SAR surface wind biases depends on geometric properties of the sensor, it may be improved by using a large network of classical metocean buoys as a training dataset in the future. On the contrary, the algorithm extrapolating surface winds to higher altitudes only depends on meteorological parameters. Therefore, it can be trained with a few Lidars in one location and applied in other areas (if similar meteorological conditions are met). In Section 4, the method is tested in two areas off the Dutch coast where profiling Lidar data are available. The SAR wind speeds extrapolated at hub height are converted into a Weibull distribution, and the extractible power is obtained by simulating the

presence of a typical 10 MW wind turbine operating at 200 m. The resulting maps are presented and compared with the output of the numerical model in order to estimate the benefit of using this method compared with a state-of-the-art technique.

**6. Line 80: wasn't WRF developed at NCAR? Add a reference.**

L. 95: The WRF (Weather Research and Forecasting) non-hydrostatic meso-scale model (Skamarock et al., 2019) was run over these areas with a resolution of 1 km.

We gave the reference mentioned in WRF website 'how to cite' section:

Skamarock, W. C., J. B. Klemp, J. Dudhia, D. O. Gill, Z. Liu, J. Berner, W. Wang, J. G. Powers, M. G. Duda, D. M. Barker, and X.-Y. Huang: A Description of the Advanced Research WRF Version 4. NCAR Tech. Note NCAR/TN-556+STR, 145 pp., doi:10.5065/1dfh-6p97, 2019.

**7. Please add a reference for GFS, too.**

No reference could be found on GFS website. We can only say it was developed by NCEP.

L. 98: It was fueled at its boundary limits by a larger-scale model, the reanalyzed GFS (Global Forecast System) having a resolution of 0.5° developed by NCEP (National Centers for Environmental Prediction).

**8. Did you run WRF in chunks (e.g., separate runs every month) or as a single run?**

The WRF was run in chunks of 24h.

**9. Figure 1: please make the fonts bigger (in the axes and the colorbar). Also, specify in the caption what the considered time period is. Also, please specify if latitude is N or S, and if longitude is E or W.**

[Figure]

**Figure 1: Locations of Zone 1 (bottom, latitude 51.50°N - 52.09°N / longitude 2.82°E - 3.77°E) and Zone 2 (top, latitude 52.15°N - 52.74°N / longitude 3.71°E - 4.68°E) with the positions of the profiling Lidars. The colour represents the number of Sentinel-1 SAR Level 2 wind observations during years 2017, 2018 and 2019.**

**10. Section 2.1: which planetary boundary layer scheme did you use in WRF? Please clarify.**

We used a specific planetary boundary layer scheme adapted to coastal environments. Although, as explained previously, we cannot provide a full list of the parameters, we provided the reference that was used to do it.

L. 96: The Planetary Boundary Layer (PBL) parametrization of the model was based on Hahmann etal., 2020.

Hahmann, A. N., Sīle, T., Witha, B., Davis, N. N., Dörenkämper, M., Ezber, Y., García-Bustamante, E., González-Rouco, J. F., Navarro, J., Olsen, B. T., & Söderberg, S.: The making of the New European Wind Atlas – Part 1: Model sensitivity. Geoscientific Model Development, 13(10), 5053-5078, doi: 10.5194/gmd-13-5053-2020, 2020.

**11. Throughout the paper, please consider using a different color scheme. The rainbow color has quite a lot of issues, see for example https://www.climate-labbook.ac.uk/2014/end-of-the-rainbow/**

We used Viridis colorbar.

**12. Line 93: again, lidars are not in-situ instruments.**

See answer to point 4.

**13. L. 95: please provide more details on the QC – is there a reference?**

There is no reference.

L. 116: The data were quality checked by our data provider C2WIND (for each time intervals, the minimum number of packets was set at 20 and the minimum availability at 80%).

**14. L. 100: if two lidars were not used in the analysis, I guess there is no need to include them in the map and in the text.**

L. 111: The dataset used in this study comprises five ground-based profiling Lidars located off the Dutch coast (Figure 1).

**15. Table 1: "lowest" and "highest" instead of "first" and "last" when talking about altitudes.**

| Lidar | Longitude | Latitude | First date | Last date | Number of levels | Lowest altitude | Highest altitude |
|-------|-----------|----------|------------|-----------|------------------|-----------------|------------------|
| HKZA | 4.011°E | 52.309°N | 2016-06-05 | 2018-06-05 | 11 | 30m | 200m |
| HKZB | 4.013°E | 52.292°N | 2016-06-05 | 2018-06-05 | 11 | 30m | 200m |
| LEG | 3.667°E | 51.917°N | 2014-11-17 | 2017-03-31 | 10 | 61m | 300m |

| EPL | 3.276°E | 51.998°N | 2016-05-30 | 2017-03-31 | 11 | 61m | 290m |
| BWFZ01 | 3.033°E | 51.71°N | 2015-06-11 | 2017-02-27 | 10 | 30m | 200m |

**Table 1: Main characteristics of the five profiling lidars**

**16. Figure 2: typo on the y-axis label. Also, what do the shading of the dots represent?**

The shading of the dote represent the density of dots that is sometimes difficult to see on scatter plots

[Figure]

**Figure 2: Exponent of the power law between the wind speeds at 4 m and 40 m as a function of the air-sea temperature difference fitted with a second-degree polynomial fit (red curve) with the following coefficients: $Y = 0.1137 + 0.0178\,X + 0.001\,X^2$. The colours represent the density of points.**

**17. Line 115: what do you mean by "correct on average"? Please provide quantitative assessments.**

Ok. 'Correct on average' means that is the exponent 0.11 is unbiased (the mean error is 0). However, we decided to refine it depending on the atmospheric stability because it can fluctuate importantly between stable and unstable conditions.

L. 130: Hsu et al. (1994) recommend choosing an exponent of 0.11 over the sea. We checked this hypothesis with HKZA and HKZB Lidars that were equipped with anemometers measuring wind speed at 4 m a.s.l.. This exponent was found to be indeed correct on average. However, in order to refine the wind speed values extrapolated at 10 m a.s.l., we adapted the exponent depending on the current atmospheric stability.

**18. L. 147: what do you mean by "around 5 AM or 5 PM"? Please be specific and provide a plot/histogram/table if necessary.**

We do not think a histogram would be necessary here.

L. 152: The revisit rate is one passage every two days, which occurs usually in the morning around 5 AM or in the evening around 5 PM (UTC). The satellites pass in the morning or in the evening depending on the orbit orientation, descending or ascending, respectively. The exact acquisition time can vary by plus or minus 30 mn depending on the incidence angle under which the region of interest is observed.

**19. I don't think Figure 3 is needed. In the text, you can simply state that you use data from 2017 to 2019 because the satellite constellation was not fully operational before that. No need to insert an histogram here (if you really want, please move it to the supplementary information).**

Figure 3 was removed.

**20. What is the difference between the map in Figure 1 and what is shown in Figures 4 and 5? If it is just a matter of the years being plotted, I would suggest plotting the 2017-19 data in Figure 1 directly, so that Figures 4 and 5 can then be removed from the paper as they do not add much information.**

Figure 4 and 5 were removed.

**21. Figure 6: "number" instead of "nb". Also in the caption, please specify which methods you are using to provide more context.**

[Figure]

**Figure 3: Wind power mean absolute error in percentage as a function of the number of samples, using maximum likelihood to fit the wind Weibull pdf (orange curve), or the method of the moments (blue curve).**

**22. L. 199: how do you quantify the "error"? What does "around" mean? Please be specific: you want to make your work replicable!**

L. 213: In order to verify this, we simulated the satellites' passages over the Lidars by computing the mean wind speed and the wind power using only the Lidar measurements at the satellites' passage times. These values were compared to those obtained using all Lidar measurements at any time of day. For each Lidar, the differences were found to be below 1% and 2%, respectively, for the mean wind speed and the wind power.

**23. Figure 7: please specify the height at which wind speed is considered (y-axis label). Also, is time on the x-axis UTC or rather local time? Please specify.**

[Figure]

**Figure 4: Intra-diurnal variability of the mean wind speed at 10 m for each Lidar. The time is given in UTC, which is close to the local time since Zone 1 and Zone 2 are located near Greenwich meridian.**

**24. L. 214: please provide context as well as references for the sentence "In addition, numerical model outputs are not as reliable as in-situ data, especially in coastal areas.". Also, what do you mean here with in-situ data? Once again, lidars are not.**

Regarding global numerical models, there is no debate that in-situ real data are more accurate. It is also obvious that coastal processes involve smaller scales than the ones in open seas, thus increasing potential errors due to their scale truncation. This is why high-resolution models like WRF are usually used to simulate coastal processes.

L. 229: In addition, GMFs were empirically designed using global numerical models as a reference, but they are not as reliable as real data, especially in coastal areas.

**25. L. 222: from the text, it is not immediately clear the purpose of these two ML models. This becomes clear later in the section, but please state that here too.**

L. 236: Given the complex relation between the sea state and the wind speed, and the number of factors able to influence it, machine learning was found to be an appropriate technique to improve the accuracy of SAR surface wind speeds and remove their biases.

**26.L.224: you need to define what "error" means for you with a precise statistical metric. Bias? RMSE? R2? Or…?**

There seems to be a misunderstanding between us and the Anonymous Referee about the definition of 'errors' and 'bias', maybe due to different versions of English language. For us the errors of the SAR are simply defined as the instantaneous errors compared to Lidar measurements, and the bias is the mean of these errors.

L. 245: In order to select the input parameters, we made a list of interesting parameters and looked for the ones related to the differences between SAR and Lidars. This was done visually by plotting scatterplots of these parameters against the errors of the SAR compared to Lidar measurements.

**27.L. 225: what do you mean by "such a correlation"? Please provide a threshold or a quantitative measure of what you did.**

We did not use a criterion, but a simple visual observation of the scatter plots. We evaluated the parameter importance, so we checked the accuracy of our choices afterwards and a visual inspection is therefore enough here.

L. 246: This was done visually by plotting scatterplots of these parameters against the errors of the SAR compared to Lidar measurements.

L. 253: The relative importance of these parameters was measured after the training stage using the feature_importances_ attribute of Scikit-learn Python toolbox (Figure 5).

**28.L. 226: please clarify what the azimuth angle, the incidence angle, the elevation angle, the backscatter are for a reader not familiar with satellite data retrievals.**

L. 247: The following parameters were selected: the SAR surface wind, the SAR wind direction, the azimuth angle (i.e., the angle between the North and the satellite track), the incidence angle (i.e., the angle between the radar illumination and the zenith of the target), the elevation angle (i.e., the angle between the radar illumination and the nadir of the satellite),…

**29.Figures 8, 11, 13: please label the x-axis with actual names (rotated to get enough space), not numbers.**

[Figure]

[Figure]

**30. L. 239: how was the test set built? Was a random half of the data, or…? At all sites, or…?**

L. 241: The algorithm was trained with the wind measured by the Lidars extrapolated to 10 m (the first Lidar level was extrapolated to this altitude with a power law, see Section 2.2). Combining all measurement sites, more than 1000 collocated data points between the Lidars and Sentinel-1 SAR could be found. The algorithm was trained with 50% of the data points randomly chosen, and the rest of them were used as a test dataset.

**31. L. 240: please provide additional details on the machine learning models and their training. Did you use cross-validation? What hyperparameters did you set? What is the structure of the neural network chosen? Did you train the model at all sites together, or at one site at a time?**

We dropped neural networks in order to simplify the paper and because it is well known that Random Forest usually performs better than them in regression tasks.

L. 238: We used a Random Forest algorithm (Breiman, 2001), which is known to perform well in regression tasks. It was implemented with the RandomForestRegressor function of Scikit-learn Python toolbox and its architecture was chosen by using cross-validation. The default hyperparameters were found to be the most appropriate ones, except the number of trees set to 240 and the maximum depth set to 20. The algorithm was trained with the wind measured by the Lidars extrapolated to 10 m (the first Lidar level was extrapolated to this altitude with a power law, see Section 2.2). Combining all measurement sites, more than 1000 collocated data points between the Lidars and Sentinel-1 SAR could be found. The algorithm was trained with 50% of the data points randomly chosen, and the rest of them were used as a test dataset.

**32. L. 240: are you talking about mean bias? Please clarify.**

We use the term 'bias' as it is commonly used, meaning a systematic error (the mean of the errors). See https://en.wikipedia.org/wiki/Bias_(statistics). For us, 'mean bias' would not be correct.

**33.Figure 9: again, specify what the different colors for the dots means. Also, larger fonts please. Also, is this for the test set only? At all sites? Please clarify.**

[Figure]

**Figure 6: Scatterplots between the SAR and Lidar wind speeds at 10 m before machine learning (a) and after machine learning (b) using the test dataset. The colours represent the density of points. The black curve is the identity line and the red curve a fourth-degree polynomial fit illustrating the bias.**

**34.Figure 10: "error" is too vague. Do you mean bias? Also, I would suggest mobbing this figure to the supplement, as all the information in it is already included in Figure 9, and it is not discussed in detail in the main text.**

Figure 10 was removed

**35.Section 3.2: the first paragraph could be moved to the introduction.**

L. 58: Regarding the extrapolation of surface wind speeds to higher altitudes, the statistical theory of turbulence provides theoretical wind profiles (see, e.g., Grachev and Fairall, 1996). However, the problem has not been satisfactorily resolved and becomes increasingly critical as the typical height of windmills increases. Empirical evidence from offshore meteorological masts measurements suggests that a simple power law could be sufficient to model the wind profile (Hsu et al., 1994). Nevertheless, the analysis of Lidar data shows that, above 40 m, the power law is no longer accurate. This limitation has led some authors to use numerical models to improve the extrapolation to higher altitudes (Badger et al., 2016). The advantage of numerical models is that they provide information about atmospheric stability through parameters like surface temperature and surface heat flux. In Badger et al. (2016), these surface parameters were averaged and combined with the similarity theory of Monin-Obukhov to extrapolate wind Weibull parameters. However, to our knowledge, this method was validated with only one meteorological mast in the Baltic Sea and not higher than an altitude of 100 m. Therefore, more research is needed to improve the estimation of wind resources at hub height with SAR data, and convince the industry to use them.

**36.Section 3.2: once again, more details are needed to fully understand how the machine learning approach was applied. See my other specific comment above for specific questions that need to be addressed.**

L. 286: The algorithm was also implemented with the RandomForestRegressor function of Scikit-learn Python toolbox. We used the default hyperparameters, except the number of trees set to 340, the maximum depth set to 50 and the maximum number of features set to 'sqrt'. The relative importance of the parameters after the training phase is shown in Figure 7.

**37.Figure 12: the y-axis label is not clear to me.**

[Figure]

**Figure 8: Bias of the extrapolated SAR wind speed against each Lidar in percentage.**

**38.Do you have any explanations on why for some lidars in Fig. 12 the performance decreases with height, while for some others it actually increases?**

Actually, the performance does not increase or decrease with altitude depending on the lidar. The y axis is the bias in %, which can be positive or negative. The dispersion that can be observed when the altitude is increasing is just the error bar of our method that slightly increases with altitude, as it could be expected: when the altitude increases, the bias of our method for a given lidar can be found in a larger interval around 0. We do not have a specific explanation on why the bias is positive, negative, going upward or downward depending on the Lidar.

**39.Figure 15: please correct the y-axis label.**

Figure 14 was removed (we assume you were talking about Figure 14)

**40.Section 4.1: it is not clear to me how the correction is performed. Please provide additional details to make your work replicable.**

See the answer to major comment 2.

L. 316: In order to compensate for this effect, we reintroduced artificially the original variability of the data. This was done by analysing the distribution of the SAR wind speed errors compared to Lidar measurements and adding a similar random variable to the SAR wind speed obtained after machine learning. The appropriate random variable was found to be a Gaussian with the standard deviation of the SAR wind speed errors. For each data point, at least five additional artificial datapoints needed to be created for the wind speed standard deviation to converge. After this bootstrap, the wind speed standard deviation error was 1.5% when considering all Lidars together in the test dataset. Thus, the result of this correction is an almost unbiased estimation of the wind speed standard deviation.

**41. In Section 4.2, you state that "Due to the short distances between the Lidars used in this study, such a validation could not be realized here.". To me, lidars that are about 100 km from each other would still allow for a validation using one for training and another for testing**

The maximum distance between the lidars is about 80 km. This would be enough on land, but on the seas, we have reservations because of the homogeneity of the sea surface and of the wind field. Moreover, there is the risk that our results could be more related to the different types of lidars that were used, rather than the geographic effect. We hope to publish soon a test of the method over an area located in another sea or ocean.

**42. Figg. 16, 17: the y-axis labels are not specific enough.**

[Figure]

Figure 10: SAR wind power error in % compared to the one computed with Lidars measurements.

**43. You can combine Figures 18 and 19 into a single one with two panels. Same for 20-21 and 22-23, and 24-25.**

[Figure]

**Figure 11: Extractible wind power over Zone 1 in kW for a typical 10 MW turbine predicted by the numerical model (a) and SAR satellites (b), difference in percentage (c), and percentage of low-quality SAR data (d).**

[Figure]

**Figure 12: Extractible wind power over Zone 2 in kW for a typical 10 MW turbine predicted by the numerical model (a) and SAR satellites (b), difference in percentage (c), and percentage of low-quality SAR data (d).**

**44. A data or code & data availability statement is missing.**

The data are available online (SAR image from ESA and Lidar data from RVO). However, due to the corporate constraints we have, unfortunately, we cannot provide our code.

**45. A conflict of interest statement is missing**

L. 390: **Author contribution**

Louis de Montera designed the algorithm and wrote the paper, Henrick Berger processed the SAR raw data and created a Level 2 gridded wind product. Romain Husson provided his expertise on SAR satellite and wind measurement from space. Pascal Appelghem parametrized the WRF model and performed the runs. Laurent Guerlou and Mauricio Fragoso supervised the study, organised the funding, and gathered together the project team.

**Competing interests**

The authors declare that they have no conflict of interest.

**Response to Anonymous Referee 2**

**Review of manuscript "High-resolution offshore wind resource assessment at turbine hub height with Sentinel-1 SAR data machine learning" by Louis de Montera et al.**

**As I am providing the second review of the manuscript, and have had the change to go through the first reviewer's comments (referred to as RC1), I can start with pointing out that I fully agree with the raised criticism and recommended changes for an improvement of the manuscript.**

**Below I list my own major and minor comments, some of them being a repetition of those in RC1 but also including some additional input:**

**(comments in order of appearance in manuscript)**

**[l 11] When referring to "Lidar measurements", first time here in the abstract, please specify what kind of measurements you mean explicitly – e.g. Doppler wind lidar, somewhat ground-based, measurements of wind velocity profile in range relevant for wind energy applications, or similar.ll Please check the overall manuscript for a sufficiently specific terminology with this respect.**

L. 10: The method is tested in two 70 km x 70 km areas off the Dutch coast where measurements from Doppler wind Lidars installed at the sea surface are available and can be used as a reference.

**[ll 13-14] When stating a bias, you also need to mention the considered reference – please add this here.**

L. 13: The SAR wind speed bias against Lidar measurements at 10 m above sea level is reduced from -0.42 m s$^{-1}$ to 0.02 m s$^{-1}$, and its standard deviation from 1.41 m s$^{-1}$ to 0.98 m s$^{-1}$.

**[ll 19-20] When reading the sentence "The accuracy of the wind power…" it becomes not clear how you get to the numbers you compare. You should also refer to the process of deriving a power curve from (any) wind data, i.e. the involved derivation.**

L. 17: Once the wind speed at turbine hub height is obtained, we assume the presence of an 10 MW turbine with a typical power curve. The extractible wind power is calculated by obtaining the wind speed Weibull distribution with the method of the moments, and then multiplying it by the turbine power curve.

**I am also wondering if you really need to consider this (as I understood later, very simplified power curved derivation) for your study, or instead could focus on a derivation of wind power density.**

The problem with deriving a total wind power density is that we would have a higher error when comparing with Lidars. When multiplying the Weibull pdf by a power curve, which is what is done in practice, the error is much lower. This is why we chose to use a power curve. More precisely, the

error is higher when computing the total wind power density, because one has to consider very low wind speeds and very high winds speeds. In these ranges, wind speed is difficult to measure precisely with SAR satellites. Our opinion is that, in these ranges, the turbine is usually not functioning anyway, so that there is no need to consider them in practice when looking at the extractible power. So, to sum up, we chose this approach because it is closer to industrial applications and because it is more appropriate to test the use of SAR satellites. Note that we now use a more realistic power curve (the one of DTU 10MW reference turbine).

**[l 39] I believe you should refer here to ground- (or bottom-) based Lidars.**

L. 42: Contrary to ground-based Lidars, spaceborne sensors have the advantage of sounding large areas with high spatial resolution.

**[ll 43-44] The sentence seems incomplete – add "… of meteorological conditions [that may impact this extrapolation]", or similar.**

L. 46: Moreover, the extrapolation of their measurements from the sea surface to hub height is not an easy task due to the variety of meteorological conditions that may impact the wind speed extrapolation ratio.

**[l 59] Be more specific here: "found to give good results" for what explicitly?**

L. 70: Regarding their extrapolation at higher altitudes, on land, machine learning has also been found to improve the accuracy of wind speeds extrapolated at turbine hub height compared to power laws or logarithmic laws (Türkan et al., 2016; Mohandes and Rehman, 2018; Vassallo et al., 2019). Optis et al. (2021) also found that machine learning was more efficient at extrapolating offshore winds than theoretical approaches.

**[l 89] Here and at other places where you introduce already available models/methods, please add a reference – in this case, for WRF.**

L. 95: The WRF (Weather Research and Forecasting) non-hydrostatic meso-scale model (Skamarock et al., 2019) was run over these areas with a resolution of 1 km. The Planetary Boundary Layer (PBL) parametrization of the model was based on Hahmann etal., 2020.

**[Figure 1] I suggest to add a larger map to help locating this cut-out. Please also introduce the used abbreviations.**

[Figure]

**Figure 1: Locations of Zone 1 (bottom, latitude 51.50°N - 52.09°N / longitude 2.82°E - 3.77°E) and Zone 2 (top, latitude 52.15°N - 52.74°N / longitude 3.71°E - 4.68°E) with the positions of the profiling Lidars. The colour represents the number of Sentinel-1 SAR Level 2 wind observations during years 2017, 2018 and 2019.**

L. 112: HKZ stand for Hollandse Kust Zuid wind farm, BWF for Borssele Wind Farm Zone, EPL for European Platform, and LEG for Lichteiland Goeree platform.

**[Figure 1 and Table 1] Are you sure that all these datasets are from floating lidars? I have at least some doubt with respect to LEG and IJM. Please re-confirm.**

After checking, we confirm LEG, EPL and IJM are not floating Lidars but actually on platforms.

L. 114: Lidars HKZA, HKZB, BWFZ01 are floating. Lidars EPL and LEG are installed on platforms.

**[l 115] From own experiences and also in line with available recommended practices, I would not fully trust the lower height (as 4 m a.s.l.) measurements from floating lidar systems – mostly from in-situ sensors heavily influenced by the structure itself. Please have some thoughts on this, and possibly consider some added uncertainty.**

We decided to use the first level of Lidar profiles and extrapolate it to 10m asl. The power law used to do that which takes into account the atmospheric stability is derived by using the anemometer at the base of the lidars. This might indeed introduce some uncertainty. However, this is still more precise than using a simple power law with a constant exponent. In our case, the uncertainty that is introduced is compensated by the second machine learning algorithm doing the extrapolation. Since the results we provide are given after applying this second algorithm trained with the Lidar

measurements at 200m, there is no need to add an additional uncertainty. In the future, we will be using the NDBC buoy network to perform the correction of SAR surface wind speeds, which will avoid this problem and generalize the method to other location.

L. 136: The anemometers located at the base of the Lidars at 4 m a.s.l. do not have a high precision and may add some uncertainty, however, since the final machine learning algorithm presented in this study is trained with Lidar measurements at hub height, this uncertainty is included in our results.

**[Figure 3] This figure is not very informative – please review the design and information included. Overall, I think you can and should reduce the number of figures in the manuscript, possibly combining some of them.**

Figure 3, 4, 5 were removed

**[Figure 4 and Figure 5] In these plots it may be helpful to have the coastline (or something else) as reference.**

Figure 3, 4, 5 were removed

**[section 2.4] As pointed out above, I think, the used power curve is too simplified. I would suggest to either apply a more realistic power curve, or instead consider another quantity as e.g. wind power density for this investigation.**

We used a more realistic power curve:

https://nrel.github.io/turbine-models/DTU_10MW_178_RWT_v1.html

As explained above, we prefer not to compute the wind power density instead since the SAR is less precise in measuring low and high wind speeds.

L. 176: We chose to simulate an 10MW turbine with a typical power curve: the DTU 10 MW Reference Wind Turbine V1 (see DTU Wind Energy, 2017, and https://github.com/NREL/turbine-models/blob/master/Offshore/DTU_10MW_178_RWT_v1.csv, last accessed September 2, 2021).

**[Figure 6] I do not think that this figure is really needed, instead you could add more details in the text.**

Actually, this figure is needed because it shows that the error of the method is highly related to the number of SAR samples. Since the number of samples is growing with time, and since it is possible to use also other SAR satellites (like ENVISAT or RADARSAT2), this figure show that it is possible to improve the accuracy of this method.

**[section 2.5] I am confused by your mentioning of "one passage every two days" and "passage times are separated by 12 h" – please be more specific here.**

Yes, this is confusing. The satellite passes every two days, and this can occur at specific times: or in the morning or in the evening. The two possible time have a difference of 12h.

L. 152: The revisit rate is one passage every two days, which occurs usually in the morning around 5 AM or in the evening around 5 PM (UTC). The satellites pass in the morning or in the evening depending on the orbit orientation, descending or ascending, respectively. The exact acquisition time can vary by plus or minus 30 mn depending on the incidence angle under which the region of interest is observed.

L. 209: Therefore, since the satellites pass at two possible times of the day separated by 12 h, according to the Nyquist-Shannon sampling theorem, they should be able to capture the majority of the intra-day variability.

**[l 222] As already stated above, please add reference for the applied methods – here the "two types of machine learning regressor[s]".**

L. 237: We used a Random Forest algorithm (Breiman, 2001), which is known to perform well in regression tasks. It was implemented with the RandomForestRegressor function of Scikit-learn Python toolbox and its architecture was chosen by using cross-validation.

**[l 229] Please specify how "the relative importance" is defined and derived.**

The relative importance is based on the mean decrease in impurity (over the trees).

L. 253: The relative importance of these parameters was measured after the training stage using the feature_importances_ attribute of Scikit-learn Python toolbox (Figure 5).

**[l 239] Also the statement "Random forest was found to outperform neural networks" needs more explanation / details.**

The reference to neural networks was dropped since it does not bring any interesting insight to the reader. It is well known that Random Forest usually outperforms them for regressions tasks.

**[Figure 9] Add a legend to the plots and the details of the red curve (fitting parameters).**

We did not add that fitting parameters on Figure 9 (now 6) since the fit is just there to illustrate the bias. However, we gave the fitting parameters of Figure 2 since this fit is used to extrapolate wind speeds to other altitudes.

[Figure]

**Figure 2: Exponent of the power law between the wind speeds at 4 m and 40 m as a function of the air-sea temperature difference fitted with a second-degree polynomial fit (red curve) with the following coefficients: $Y = 0.1137 + 0.0178\,X + 0.001\,X^2$. The colours represent the density of points.**

**[l 266] Again, "machine learning" needs more explanation and details.**

L. 283: These parameters were used together with the corrected SAR wind speeds at 10 m as input to the Random Forest algorithm, which was trained to learn the Lidar wind speed at several altitude levels until 200 m using the same training dataset as previously. The algorithm was also implemented with the RandomForestRegressor function of Scikit-learn Python toolbox. We used the default hyperparameters, except the number of trees set to 340, the maximum depth set to 50 and the maximum number of features set to 'sqrt'. The relative importance of the parameters after the training phase is shown in Figure 7.

**[Figure 12 (and Figure 14)] It is not clear to me, why you have not used a smaller range for the vertical axis – please revise.**

We used a -6% to +6% axis.

[Figure]

**Figure 8: Bias of the extrapolated SAR wind speed against each Lidar in percentage.**

[Figure]

**Figure 10: SAR wind power error in % compared to the one computed with Lidars measurements.**

**[l 323] Sentence "In this case, …" is incomplete.**

The word 'the distribution' was missing.

L. 323: In this case, the distribution was shifted to the left, which means that the numerical model underestimates the wind speed compared to Lidar measurements.

**[Figure 18 and following] Please re-arrange these plots for better comparability – combine several plots in one figure, for instance.**

[Figure]

**Figure 11: Extractible wind power over Zone 1 in kW for a typical 10 MW turbine predicted by the numerical model (a) and SAR satellites (b), difference in percentage (c), and percentage of low-quality SAR data (d).**

[Figure]

**Figure 12: Extractible wind power over Zone 2 in kW for a typical 10 MW turbine predicted by the numerical model (a) and SAR satellites (b), difference in percentage (c), and percentage of low-quality SAR data (d).**

---

## Referee Report (RR1)

**High-resolution offshore wind resource assessment at turbine hub height with Sentinel-1 SAR data and machine learning**

*Louis de Montera, Henrick Berger, Romain Husson, Pascal Appelghem, Laurent Guerlou, Mauricio Fragoso*

**REVIEW – round 2**

**GENERAL COMMENT:**

The authors addressed most but not all of my previous comments. Despite the modifications to the approach, I feel there are still some fatal flaws (and several other minor issues), as I will detail in my comments below.

Please note that I have reviewed up to Section 3, as I really need to see my major comment on the methodology addressed before I can evaluate the results of the analysis, should the authors still want to pursue publication of this work.

**MAJOR COMMENTS:**

1. I still cannot see any practical application for the approach the authors proposed in the paper.
   In the first part of the approach (described in Section 3.1), you 1) extrapolate lidar winds down to 10 m and 2) apply a machine learning model where the SAR-derived wind speed is an input, and the target variable is the 10-m lidar wind speeds. You then use this SAR-corrected winds for the extrapolation to hub-height.
   Now, to me this approach has two fatal flows.
   The first fatal flaw I am seeing is that if one needs to have lidar data available (you use them to correct the SAR winds, Section 3.1), why would one need to extrapolate SAR data in the first place, since the lidar provides hub-height winds already?
   To respond to this concern, I could see an application of this approach when someone only has let's say a 10-m sonic anemometer (whose wind speed observations are used to correct the 10-m SAR winds), without any hub-height observations. But for this application to be possible, the authors would need to test the generalization of the approach they propose with a round-robin validation. In other words, the authors would need to answer the following question: "how accurate is this whole approach when applied at a site (i.e., in my example, where I only have near-surface wind speed observations) different from the one where it has been trained (i.e., in my example, where I have lidar observations which already give me hub-height data)?". Since the authors have multiple observational locations, they could do this exercise, but currently this validation is not done in the paper.
   However, even if this validation exercise were to be completed, the second fatal flow I am seeing here would still kick in. If one needs to have *any* 10-m observation

of wind speed to correct for the SAR data, why would one use the SAR data in the first place, instead of just extrapolating to hub-height the 10-m observations coming from the instruments needed to correct the SAR data?

**SPECIFIC COMMENTS:**

1. L. 33: other data sources can be used to estimate hub-height winds, for example reanalysis products.
2. L. 62: "Nevertheless, the analysis of Lidar data shows that, above 40 m, the power law is no longer accurate." is a strong and very general statement. While it has some merit, it needs references.
3. Section 2.1 have multiple instances of weird spacing between words.
4. L. 135: why did you consider two lidars only to determine the exponents of the power law, which are then applied to all the lidars?
5. L.240: "The default hyperparameters were found to be the most appropriate ones". How did you find this? Remember, your work should be replicable! A similar comment applies to line 286.
6. Figure 8: the y-axis label can simply be "SAR – lidar wind speed bias (%)"
7. Despite my previous comment, a data or code & data availability statement is still missing. You should add one even if your code cannot be shared – simply state it.

---

## Referee Report (RR2)

[referee-annotated manuscript omitted]

---

## Author Response (AR2)

**Revised version of manuscript WES-2021-35 and answers to comments**

**Editor's letter:**

Dear Authors,

The two reviewers raised several critical issues, and one opted for "reject". Hence, it is clear that the revised manuscript needs a new substantial revision to become publishable. The most crucial issue is your proposed method's applicability and scientific soundness. In addition, it seems you have not satisfactorily responded to all the comments of the first revision round.

I think the paper still has some valuable aspects. So, I suggest that you submit a detailed plan of the changes you propose to make and respond to all the issues from the previous and past round of comments, and we will continue with the review process. If you cannot comply with the requested changes, you can withdraw your submission.

Best regards,
Andrea Hahmann

**Response:**

Dear Editor,

First of all, we would like to thank you for accepting the management of this review process.

Concerning the paper itself, we would like to say that we did answer all the comments from the first revision round, and especially the major ones. You can have a look to the documents to confirm this. The so-called 'critical flaws' pointed out by Referee1 are actually new major comments from him or her. Therefore, Referee1 advice to reject the paper is not related to a supposed lack of response regarding the first round of review.

Actually, our paper is a major breakthrough since it is the first time SAR can be used to estimate wind resource at hub height with a sufficient accuracy to convince the industry. Moreover, the paper presents a stand-alone method to correct SAR surface winds and another stand-alone method to extrapolate surface winds to higher altitudes. Therefore, we definitely want this paper to be published and we propose you a revised version of the manuscript.

Since Referee1 seems not to understand that the first version of the paper was a proof of concept, and encourages us to disclose our operational method, we included several important improvements in the revised version, which are explicitly described in the paper and listed below. Please also consider that we were originally aiming at proposing a second paper describing an operational version of this first proof-of-concept methodology. Given the time we had between the first and the second revision, we now propose to directly describe the operational methodology we have reach, thereby also addressing the major comment from Referee 1.

Additional improvements with respect to previous version:

- The SAR data are now produced with CMOD7 GMF instead of CMOD5n GMF

- The correction of SAR surface wind is done with a network of buoys located in the US

- The correction now uses also as input the cross-polarization backscatter (it improves strong winds retrieval)

- The correction now uses also as input the ECMWF wind speed provided as SAR metadata (it improves low winds retrieval)

- The WRF is now forced with ERA5 1h instead of GFS 3h (it is more adapted to the small areas of study)

- We now use a Gradient Boosting algorithm instead of Random Forest (it is more efficient and does not disrupt the wind speed distribution)

- The validation of the extrapolation is done with a round-robin technique (so that all samples for each Lidar can be used to estimate wind power)

- The relative importance of parameters is computed with ShAP method.

- The error due to SAR low and irregular sampling is corrected exactly and automatically by using the WRF to simulate the satellites' passages

We would like to emphasise that we reject Referee1 statement that our paper cannot be published because our method is not operational. Firstly, the immediate applicability of scientific results has never been a requirement to publish in scientific journals. Secondly, the algorithm now presented in the revised version is fully operational and was recently used by the French Government to assess AO4 and AO5 offshore sites respectively located in Normandy and Southern Brittany.

We believe that the review from Referee1 is unfair and may raise the question of his or her partiality. For instance, we have found that a team that we recommended as referee for our manuscript was planning to do exactly the same study as ours (see the numerous references about future plans of using random forest to extrapolate SAR winds in Optis et al.: New methods to improve the vertical extrapolation of near-surface offshore wind speeds, Wind Energ. Sci. Discuss. 2021 https://wes.copernicus.org/articles/6/935/2021/). Therefore, we suspect that Refeee1 may have a personal interest in preventing or delaying the publication of our results. Due to this context and if this was the case, the reviewing process may require a conflict-free reviewer, or at least some careful discernment from the editor with respect to the referee's review.

Finally, we would like to thank you again for your time and expertise.

Best regards,

The authors

**Referee1's comments:**

MAJOR COMMENTS:

1. I still cannot see any practical application for the approach the authors proposed in the paper. In the first part of the approach (described in Section 3.1), you 1) extrapolate lidar winds down to 10 m and 2) apply a machine learning model where the SAR-derived wind speed is an input, and the target variable is the 10-m lidar wind speeds. You then use this SAR-corrected winds for the extrapolation to hub-height. Now, to me this approach has two fatal flows. The first fatal flaw I am seeing is that if one needs to have lidar data available (you use them to correct the SAR winds, Section 3.1), why would one need to extrapolate SAR data in the first place, since the lidar provides hub-height winds already? To respond to this concern, I could see an application of this approach when someone only has let's say a 10-m sonic anemometer (whose wind speed observations are used to correct the 10-m SAR winds), without any hub-height observations. But for this application to be possible, the authors would need to test the generalization of the approach they propose with a round-robin validation. In other words, the authors would need to answer the following question: "how accurate is this whole approach when applied at a site (i.e., in my example, where I only have near-surface wind speed observations) different from the one where it has been trained (i.e., in my example, where I have lidar observations which already give me hub-height data)?". Since the authors have multiple observational locations, they could do this exercise, but currently this validation is not done in the paper. However, even if this validation exercise were to be completed, the second fatal flow I am seeing here would still kick in. If one needs to have any 10-m observation of wind speed to correct for the SAR data, why would one use the SAR data in the first place, instead of just extrapolating to hub-height the 10-m observations coming from the instruments needed to correct the SAR data?

> The first version of the paper was a proof of concept, not yet as an operational product, and we explicitly mentioned in it that we planned to use a buoy network in the future. So we do not see it as a 'critical flaw', but rather as a logical step of our research and development. In any case, since the last round of review the algorithm became fully operational, so we take advantage of this revised version to introduce a correction of SAR surface winds with the NDBC buoy network.

> For what concerns what you call the "second fatal flow", we recall that the purpose of the in situ calibration at the sea surface is to be able to correct for systematic errors that are not site specific. They aim at correcting errors related to the GMF and to the sensor calibration. They can therefore be learnt from measurements located elsewhere and at different periods. There is no need to have this source at the location and period of interest, which widens the potential for using it in various locations.

> Regarding the extrapolation, we now do a round-robin validation for the sake of increasing the number of sample per Lidar used in wind power estimation. As already explained in the first round of review (in our answer to minor comments), we do not think that the round-robin method adds a lot of value here in terms of validation because the meteorological conditions of the samples are already not correlated (48h minimum time difference) and because the Lidars are too close form each other. In any case, it confirms directly that the method can be trained in one place and applied in another.

SPECIFIC COMMENTS:

1. L. 33: other data sources can be used to estimate hub-height winds, for example reanalysis products.

> That's what we mean by 'numerical models'.

2. L. 62: "Nevertheless, the analysis of Lidar data shows that, above 40 m, the power law is no longer accurate." is a strong and very general statement. While it has some merit, it needs references.

> We cite in the next paragraph a lot of literature trying to find a more accurate model to extrapolate to hub height. We also checked this with our own Lidar data and confirm the power law is not suitable after 40m. We added a reference (Tieo et al., 2020)

> *L. 69: 'Nevertheless, above a few tens of meters, the power law model is no longer accurate (see, e.g., Tieo et al., 2020). This limitation has led some authors to use numerical models outputs to improve the extrapolation to higher altitudes (Badger et al., 2016).'*

3. Section 2.1 have multiple instances of weird spacing between words.

> Ok. Corrected.

4. L. 135: why did you consider two lidars only to determine the exponents of the power law, which are then applied to all the lidars?

> It was explained in the previous version of the paper that only these two Lidars had an anemometer at their basis. Anyway, this is not needed in the revised version since we now use the NDBC network and do not extrapolate the Lidar data to a lower altitude.

5. L.240: "The default hyperparameters were found to be the most appropriate ones". How did you find this? Remember, your work should be replicable! A similar comment applies to line 286.

> We always used Gridsearch (with cross-validation). We mention now it explicitly in the revised version.

> *L.310: 'The Gradient Boosting hyper-parameters optimized with grid-search are shown in Table 3 (left column). The other hyper-parameters are the default ones. The relative importance of the input parameters is given in Figure 4.'*

6. Figure 8: the y-axis label can simply be "SAR – lidar wind speed bias (%)"

> Ok

7. Despite my previous comment, a data or code & data availability statement is still missing. You should add one even if your code cannot be shared – simply state it

It is not clear if this section is mandatory in WES guidelines, so we simply added the data providers in the acknowledgment section in the first version. We now added this statement accordingly to your comment. Unfortunately, we are still not allowed to share the code since our company is commercial.

*Code and data availability*

*SAR data are available at ESA. Buoys data are available at NDBC. Lidar data are available at the Dutch Ministry of Economic Affairs and Climate Policy. The WRF source code and Python packages are open source. Unfortunately, the full code of the method developed in this paper is not available due to corporate constraints.*

**Referee2's comments:**

This manuscript is about vertical extrapolation of wind fields from satellite SAR. It is novel and interesting in the context of offshore wind energy projects and planning.

GENERAL COMMENTS:

1. The work is very focused on wind resource assessment and on mapping the wind power potential at the height 200 m. I think, however, that the real advantage of extrapolating instantaneous SAR wind fields lies in the possibility to map the detailed variation of instantaneous winds and compare these with e.g. numerical modeling. This aspect is not mentioned.

> Yes, we need to mention that since it is a really interesting application. Actually, there are 3 important possible applications in the paper: correction of SAR surface winds to create better products, instantaneous extrapolation as you mention, and resource assessment. It is clear from our observations that instantaneous SAR images at hub height also offer additional interesting insights.

> *L.25 The algorithms presented in this study are independent from each others and can therefore also be used in a more general context to correct SAR surface winds, extrapolate surface winds to higher altitudes, or produce instantaneous SAR wind fields at hub height.*

> *L. 415 The resulting SAR wind speed bias is 0.02 m s$^{-1}$. Its MAE is 0.57 m s$^{-1}$ and its standard deviation 0.74 m s$^{-1}$. This algorithm can be used as a standalone to create more accurate SAR wind products. The second algorithm extrapolating surface winds to higher altitudes has been tested against Lidar measurements […] This algorithm can also be used as a standalone to extrapolate wind speeds measured at 4 m above sea level. These two algorithms combined together produce instantaneous SAR wind fields at hub height, which can provide interesting insights to wind farm developers.*

2. A second advantage of the presented approach is that it overcomes the current shortcomings of analytical approaches such as MOST. These are only valid within the surface layer of the atmosphere whereas the machine learning approach can be used at any height as long as there is sufficient data available for training and testing. This aspect is not mentioned.

> Yes, it is clearly another important advantage of machine learning. We modified the introduction accordingly to stress this point.

> *L. 84 Moreover, machine learning can be used at any altitude contrary to theoretical approaches that are limited to the boundary layer.*

3. Like previous reviewers, I find that the use of a turbine specific power curve makes the analysis overly complicated and more difficult to follow. I would recommend to map the wind power density instead. If this change is not feasible at this stage, please add some reflections over the effects of using the power curve.

We already answer Referee2 about this point. SAR sensors can have difficulties in detecting very high wind speed and since these wind speeds have a very strong weight in the total power density, SAR estimations of the total power density would be less accurate. So, in order to show that SAR can be used for resource assessment, we chose to use the extractible power instead, which is more accurate since the power curve has a plateau at high wind speeds and is less affected by potential inaccuracies at strongest wind regimes. We consider using the extractible power is not a flaw since the industry computes it from the Weibull parameter in practical applications. We explain it in more details in the revised paper.

*L. 232 Since the total wind power density is related to the cube of wind speed, very high wind speeds have a strong influence on its estimation. Since SAR sensors do not detect well very high wind speeds because they tend to saturate, we do not recommend using them to estimate the total wind power density. However, estimating the extractible wind power instead removes this limitation, because wind turbines usually do not operate or function at a plateau when very high wind speeds occur.*

4. I think there is some confusion about the term 'hub height'. The 10 MW reference turbine used here has a hub height of 119 m but throughout the manuscript, the 200 m height is described as the 'hub height'.

Ok. We changed all the results to provide maps at 120m. As explained above, the idea was to use a standard power curve in order to remove low and very high wind speeds from the wind power assessment. So we were not focused on the real hub height of the turbine, but just on the shape of the power curve. Therefore, we chose the DTU 10MW turbine, and assumed that a turbine operating at 200m would have more or less the same power curve shape (multiplied by a constant). To avoid such assumption, we produced new maps at 120m.

*L. 327 Figures 10 and 11 show the extractible wind power maps at 120 m produced by the WRF and SAR methods assuming a typical 10 MW turbine, and the difference between them in percentage.*

5. The quality of the different data sets is not really considered. In particular, I would like to know more about the parameters from WRF: is the accuracy of the instantaneous temperatures and heat fluxes sufficient for this type of analysis. See for instance: Pena Diaz, A., & Hahmann, A. N. (2012). Atmospheric stability and turbulence fluxes at Horns Rev— an intercomparison of sonic, bulk and WRF model data. Wind Energy, 15(5), 717–731. https://doi.org/10.1002/we.500.

Actually, if the numerical were perfect, there would be no need for SAR and machine learning. So we totally agree that the outputs of the numerical model have a low quality. However, this is not the main issue, because the idea here is precisely to learn the errors of the numerical model through machine learning. So our approach was to remove the most unreliable model parameters (like the wind speed), and keep more reliable ones (like the relative extrapolation ratio) or the less fluctuating ones (like heat flux and temperature). By doing so, it seems that we were indeed able to learn the numerical model errors and exploit the remaining information they contain. We explained it better and cited the above-mentioned paper.

*L. 207 Since the accuracy of numerical models outputs is questionable, one must be careful when choosing these meteorological parameters. In particular, the WRF wind speed at hub height could not be used directly since the aim of this algorithm is to estimate it with SAR satellites. Instead, we provided the algorithm with the WRF extrapolation ratio between the wind speed at the sea surface and hub height. Using this relative quantity has the advantage of preventing the WRF from interfering with SAR estimates. Moreover, this extrapolation ratio was found to be accurate: the comparison with experimental data shows that its bias was less than 1% for each Lidar. The other relevant parameters related to the atmospheric stability we used were the air-sea temperature difference and the surface heat flux. The accuracy of these parameters is also problematic (see, for example, Pena Diaz & Hahmann, 2012). However, in the context of machine learning, the focus is more on the information they contain, rather than their absolute accuracy. Since they are not fluctuating as quickly as the wind speed, we assumed that their biases were following repetitive patterns that could be learnt by the algorithm, and that these biases would not prevent it from extracting the relevant information.*

6. Finally, the presentation of results seems a bit unstructured as results are spread across sections 2, 3, and 4. Sentences alternate between present and past tense. Please be consistent.

Ok, we also created a new section called 'Methods' to clearly separate the methods, data and the results. We will use only past tense.

SPECIFIC COMMENTS:

1. 'it' refers to the error? Perhaps better to state that. (L. 22)

Corrected

*L. 17 Once the wind speeds at hub height are obtained, we assume the presence of a 10 MW turbine and estimate the wind Weibull parameters taking into account the SAR irregular temporal sampling. The wind speed Weibull distribution is then multiplied point-by-point by the turbine power curve to obtain the extractible wind power with a 1 km spatial resolution.*

2. This description does not fit in here. Perhaps more suitable for Section 2. (L. 81)

Corrected. It was moved to section 3 Methods.

*L. 179 Given the complex relation between the sea state and the wind speed, and the number of factors able to influence it, machine learning was found to be an appropriate technique to improve the accuracy of SAR surface winds. Since the error depends on the geometry of the sensor, this algorithm was to be trained with a large database of measurements covering the diversity of possible angles obtained from the NDBC network of metocean buoys (Section 2.4).*

*L. 201 After this correction, the extrapolation of SAR surface winds did not depend on the sensor geometry, therefore, the algorithm could be trained with a dataset including a limited number of instruments, like the Lidar data from the North Sea (Section 2.5).*

3. This really depends on a project's level of maturity. A first screening of sites might be based on numerical modeling alone but as a project gets closer to a financing decision, in situ observations are always used, as far as I know. (L. 99)

> Yes, actually we wanted to say that the numerical model was typical of the ones used by the industry, not the whole assessment. In any case, we were already using Lidar data to correct the WRF like the industry is doing (see the previous version of the paper in the results section). In the revised version, we explain now clearly this use of in-situ instruments to correct the WRF bias in the Data section.

> *L.141 Moreover, since the WRF is typical of numerical models currently used by industry, we also used it as a reference to assess the benefits of using SAR data (Section 4). Since numerical models are often combined with in-situ measurements to increase their accuracy, we also corrected the WRF bias. The extractible power estimated by the WRF was found to be underestimated by 3% compared to Lidars.*

4. Normally, u and z are used for instantaneous observations (instead of U and Z). (L. 126)

> Corrected

5. I suggest to put the description of SAR data first - before the model and reference data sets. The SAR data represents the core of this work. (L. 146)

> Corrected. It is more logical indeed.

6. I do not think that readers of WES will know the difference between grid spacing and spatial resolution. Please explain or give the grid spacing alone. (L. 148)

> Corrected

> *L. 109 Sentinel-1 Level 1 Ground Range Detected (GRD) backscatter product has a spatial resolution of a few tens of meters, whereas Level 2 wind products typically have a spatial resolution of 1 km.*

7. I am a bit confused here: Is the reference used to produced Figure 3 based on a direct calculation of the power from the data set itself without any curve fitting)? It should be. Please add this information. (L. 179)

> The reference used to produce this Figure does not involve any curve fitting. To obtain the extractible power reference, we use the arbitrarily chosen Weibull parameters and the exact formula to get the Weibull pdf. Then we multiply it point-by-point by the 10MW turbine power curve. We explained it with more details and clarity.

> *L. 260 The accuracy of this estimation method was assessed with simulations by generating time-series of a Weibull random variable with arbitrary parameters, and then trying to recover the original parameters from these time-series. More specifically, we chose Weibull parameters typical of the North Sea wind climate ($k = 2.2$ and $\lambda = 8.5$) and computed the*

*reference extractible power using these parameters and the exact formula (Eq. (2) multiplied point-by-point by the 10 MW turbine power curve). Then, we generated random synthetic wind speed time-series using the Weibull pdf (Eq. (2)) and applied the method of the moment (Eqs. (3) and (4)) to recover the original Weibull parameters and estimated again the extractible power.*

8. Again, please specify what is meant by 'the original parameters'. (L. 194)

The original parameters are arbitrary parameters typical of the wind speed Weibull distribution we found in the area of study with the Lidar data. We used k=2.2 and lambda=8.5.

*L. 262 More specifically, we chose Weibull parameters typical of the North Sea wind climate (k = 2.2 and $\lambda$ = 8.5)*

9. This could be re-phrased. In fact, I do not think a low number of samples can be called an advantage. (L. 207)

Corrected

*L. 279 This limitation actually guarantees the statistical independence of measurements, nevertheless, since SAR satellites are on a sun-synchronous orbit, they pass always at the same times of the day, in the morning or in the evening. As a result, they cannot fully see the intraday variability of the wind.*

10. I would like to see more information about these calculations - sounds a bit too good to be true. How many samples were used and did you calculate the ME or the MAE? (L. 215)

We do not see the reason why this result 'sounds a bit too good to be true'. We actually explained the reason why the error due to the SAR temporal sampling is expected to be low: 'It can be seen that the wind diurnal cycle is close to a 24 h period sinusoid. Therefore, since the satellites pass at two possible times of the day separated by 12 h, according to the Nyquist-Shannon sampling theorem, they should be able to capture the majority of the intra-day variability.'

To be more specific, we added a reference on the Van der Hoven spectrum of the atmosphere showing that there is a spectral gap between the diurnal peak and the small-scale turbulence. Therefore a sampling with a 12h time difference should indeed catch most of the diurnal and intra-day variability:

[Figure]

Regarding the errors, 'ME or MAE' are irrelevant here because we are not dealing with time-series, but scalar values. The errors are simply the error of the mean wind speed and the error of the wind power (in absolute value and %).

We added a table with the details of the results.

*L. 283 The intraday variability of wind speed is low (Van der Hoven, 1957) and close to a 24 h period sinusoid (Figure 3). Therefore, since Sentinel-1 satellites pass at two possible times of the day separated by 12 h, according to the Nyquist-Shannon sampling theorem, it should be enough to capture the intraday variability. In order to verify this, we computed the mean wind speed and the extractible wind power using only Lidar measurements at 5 AM and 5 PM (UTC). Then, we compared results to the ones obtained using all Lidar measurements at any time of day. For all Lidar, the differences were found to be below 0.5% and 1%, respectively (Table 2).*

| Lidar | Error of the mean wind speed in % | Error of the extractible wind power error in % |
|-------|-----------------------------------|------------------------------------------------|
| HKZA | -0.34 | -0.16 |
| HKZB | -0.23 | -0.01 |
| LEG | 0.36 | 0.94 |
| EPL | -0.04 | 0.06 |
| BWFZ01 | -0.47 | -0.08 |

11. This sentence could be modified - it is not really about preventing the use of SAR data but rather about achieving the best possible accuracy on wind resource estimates. (L. 216)

Corrected.

*L. 288 Therefore, the satellites are indeed able to capture most of the wind intraday variability.*

12. Please comment on the differences between these curves somewhere in the text. Why is the diurnal variability less pronounced for HKZA and HKZB? (L. 221)

> We agree that this difference is strange, but we are able to provide an explanation because we did not produce these data and do not have enough information about the measurement campaigns. It may be due to the lack of accuracy of the Lidars' first level. In the revised version, we give the curves at 120m instead and this problem seems to disappear.

[Figure]

13. The GMFs I know of are developed through triple-collocation using both model and in situ observations from buoys. Please check the literature and reconsider this sentence. (L. 221)

> These GMF were designed mostly with ECMWF numerical model (see section 2 in Hersbach 2008 CMOD5.N: A C-band geophysical model function for equivalent neutral wind published by ECMWF). Actually, according to Stoffelen et al. 2017, the triple collocations with buoys were only used for validation purposes and a posteriori bias correction. Since this bias correction depends strongly on the considered scatterometers, in any case, we doubt it is relevant for SAR. So, we maintain our argument that the design of GMF is not well adapted to coastal areas because the ECMWF model is less accurate in these areas. However, we modified our statement to include your comment.
>
> *L. 60 Another reason is that GMFs were designed empirically using the ECMWF model as a reference, which may not be accurate in coastal areas (in-situ data were used only for validation and a posteriori bias correction, see Stoffelen et al., 2017, and references therein).*

14. What is meant by 'interesting parameters'? (L. 245)

> Corrected. We selected parameters known to be related to SAR errors due to physics o due to the retrieval algorithm design.
>
> *L.187 Regarding input parameters, we selected parameters related to SAR wind speed retrieval errors because of physics or because of the retrieval algorithm specificities.*

15. When such a statement is made, we need to see the evidence - the numbers behind. Please provide them e.g. in a table. (L. 281)

> Since we used a PBL more adapted to the higher boundary layer, we decided to verify the accuracy of the WRF surface levels. We did this by extrapolating Lidar data to lower altitudes, which is not very reliable, and found a strong bias. So we are unsure of these results and do not wish to present them, but, as a precaution, we decided to remove the WRF levels below 40m. If we had had accurate in-situ measurement below 40m, we would have given details, but we don't have any. Our aim here is just to tell the reader to be cautious with these surface levels when using a PBL adapted to higher altitudes, that we found a possible problem, but that we cannot conclude.

> *L.218 However, when assessing the WRF against Lidars, we found that the WRF wind speed had an unrealistic bias below 40m. It was unclear if this was due to the PBL adapted to higher altitudes, to a lack of accuracy of the Lidars at their first levels, or to the power law extrapolating these first levels to a lower altitude. In any case, as a precaution, we chose to use the WRF parameters at 40 m instead of the one from the surface level when producing the various input parameters.*

16. If I understand correctly, the 10-m SAR wind is first modified through machine learning to match the lidar wind speed. Next, this 10-m wind speed is extrapolated up to 200 m. Since the starting point at 10 m is identical, it is not surprising that a good match between the wind profiles is found? (L. 292)

> Separating the algorithm into two steps does not artificially improves the performance, because the training of the correction is done with the same training dataset as the as the one used to train the extrapolation. So the validation is independent and the starting point at 10m is not identical: the algorithm has to predict it before doing the extrapolation.

> In any case, in the revised paper, we now transform the 10m SAR data into the equivalent 4m buoy wind speed with an algorithm trained with the NDBC buoy network located in the US. So the datasets are clearly independent.

17. It seems like there is some confusion about the term 'hub height'. For the 10 MW reference turbine used in this study, the hub hight is 119 m. You have used 200 m, which is approximately the hub hight + blade length i.e. the maximum height of the turbine. (L. 292)

> See the answer to major comment 4 above.

18. Once again, it is difficult to follow this unless some evidence is provided in terms of numbers, tables, ... (L. 294)

> This statement was about a failed attempt, so it was neither necessary nor useful. We removed it.

19. What if the surface head flux from WRF is inaccurate as reported in the literature? How would this impact your results? (L. 296)

> See the answer to major comment 5 above.

20. I suggest to put the 'altitude' on the y-axis as it is the convention in wind energy. And call it 'height' instead. (L. 301)

Corrected.

[Figure]

21. In the previous sections, many results were presented. I suggest to restructure so all results are presented in the 'Results' section. (L. 304)

Corrected. All machine learning algorithms performance were moved to the result section.

22. Why and when was this correction performed? Should be described in the Methods section. (L. 325)

Corrected. We gave more details in the Data section. See answer to minor comment 3:

Yes, actually we wanted to say that the numerical model was typical of the ones used by the industry, not the whole assessment. In any case, we were already using Lidar data to correct the WRF like the industry is doing (see the previous version of the paper in the results section). In the revised version, we explain now clearly this use of in-situ instruments to correct the WRF bias in the Data section.

*L.141 Moreover, since the WRF is typical of numerical models currently used by industry, we also used it as a reference to assess the benefits of using SAR data (Section 4). Since numerical models are often combined with in-situ measurements to increase their accuracy, we also corrected the WRF bias. The extractible power estimated by the WRF was found to be underestimated by 3% compared to Lidars.*

23. I think, the most striking result is that the coastal wind speed gradients are resolved by SAR and not by WRF. Please elaborate on that. (L. 347)

Corrected.

L. 375 In particular, the coastal wind speed gradient, which is often crucial in offshore site assessments, is resolved by the SAR and not by the WRF (see Figure 12).

We also added a figure showing a coastal gradient on a perpendicular to the shoreline.

[Figure]

**Figure 12: Extractible wind power coastal gradient at 120 m on a horizontal line at the top of Zone 2 estimated by the WRF and by SAR satellites.**

24. This belongs to the 'Methods' section. Please elaborate on the Koch filter or at least give a reference. (L. 352)

> Actually, the use of Koch filter was already explained in the Data and Methodology section about SAR data (now 2.2) and a reference was given.

25. But mast/lidar observations are needed as well? This should be discussed in terms of the practical application of your method. (L. 380)

> The measurements used in the validation are statistically independent since they are measured with more than 48h time difference, so we do not think mast/lidar are needed to apply the method. Moreover, the revised version now uses a round-Robin validation, which shows clearly that that the method can be trained in one place and applied in another.

> However, we are not satisfied with the round-robin validation since the Lidars are too close from each other, and because it was not possible to fully test the approach in other seas due to the lack of freely accessible Lidar data. Therefore, we would like to perform more validation with more Lidars in the future.

26. Is it realistic to develop a general approach for all seas? Or will there always be a need for in situ measurements? (L. 385)

> As explained above, in-situ measurements are not needed. However, since the training of the extrapolation was done in the North Sea, we do not think that it can be applied directly in all seas. To apply the method in seas having a very different wind climate, like the Mediterranean Sea, we expect that Lidars located in the region would need to be used to train the algorithm. We clarified it in the conclusion of the paper.

> *L.430 Further research should focus on removing remaining artefacts on the SAR wind power maps, such as swath edges, bright targets, and the effect of bathymetry. Moreover, since the*

*method was validated only using Lidars located in the North Sea, the extrapolation algorithm may not be adapted to meteorological conditions in seas having a different wind climate. In that case, wind profiles measured by Lidars located in the region where the site is located would need to be included in the training dataset and used to validate the method.*

---

## Author Response (AR3)

Dear Editor, dear referees

According to your recommendations, the English was checked and improved by a professional.

Thank you very much for your time and contribution, as well as for the quality of the reviews.

Best regards,
The authors